# Harnessing the regenerative potential of *interleukin11* to enhance heart repair

Kwangdeok Shin[1], Anjelica Rodriguez-Parks [1], Chanul Kim [1], Isabella M. Silaban [1], Yu Xia[2], Jisheng Sun [3], Chenyang Dong[4], Sunduz Keles [4], Jinhu Wang [3], Jingli Cao [2] & Junsu Kang [1,5] ✉

Balancing between regenerative processes and fibrosis is crucial for heart repair, yet strategies regulating this balance remain a barrier to developing therapies. The role of Interleukin 11 (IL11) in heart regeneration remains controversial, as both regenerative and fibrotic functions have been reported. We uncovered that *il11a, an Il11* homolog in zebrafish, can trigger robust regenerative programs in zebrafish hearts, including cardiomyocytes proliferation and coronary expansion, even in the absence of injury. Notably, *il11a* induction in uninjured hearts also activates the quiescent epicardium to produce epicardial progenitor cells, which later differentiate into cardiac fibroblasts. Consequently, prolonged *il11a* induction indirectly leads to persistent fibroblast emergence, resulting in cardiac fibrosis. While deciphering the regenerative and fibrotic effects of *il11a*, we found that *il11*-dependent fibrosis, but not regeneration, is mediated through ERK activity, suggesting to potentially uncouple *il11a* dual effects on regeneration and fibrosis. To harness the *il11a*'s regenerative ability, we devised a combinatorial treatment through *il11a* induction with ERK inhibition. This approach enhances cardiomyocyte proliferation with mitigated fibrosis, achieving a balance between regenerative processes and fibrosis. Thus, we unveil the mechanistic insights into regenerative *il11* roles, offering therapeutic avenues to foster cardiac repair without exacerbating fibrosis.

Unlike adult mammals, adult zebrafish possess the remarkable capacity to regenerate lost cardiac tissues upon injury[1,2]. This regenerative ability of cardiac muscle cells can be achieved by dedifferentiation and proliferation of pre-existing cardiomyocytes (CMs)[3–5]. The regenerative events in CMs are orchestrated by molecular signals provided by activated non-muscular cells, including epicardium, endocardium, and immune cells, which create a regenerative niche[3,6–12]. Furthermore, similar to injured mammalian hearts, zebrafish exhibit an acute response to cardiac injury by depositing extracellular matrix (ECM) proteins at the injury site[13]. Although fibrosis is generally considered detrimental to regeneration, cardiac fibrosis is a double-edged event for heart regeneration as it plays a critical function to prevent rupture during the early phase of heart regeneration[13–15]. Indeed, preventing fibrosis during the early phase of the injured zebrafish hearts leads to defective heart repair[13]. In opposition to mammals, the cardiac fibrotic tissue in zebrafish hearts transiently form and are eventually replaced by newly formed CMs[16,17]. Therefore, dissecting the key signaling pathways governing

[1]Department of Cell and Regenerative Biology, School of Medicine and Public Health, University of Wisconsin - Madison, Madison, WI, USA. [2]Cardiovascular Research Institute, Department of Cell and Developmental Biology, Weill Cornell Medical College, New York, NY, USA. [3]Cardiology Division, School of Medicine, Emory University, Atlanta, GA, USA. [4]Departments of Statistics and of Biostatistics and Medical Informatics, University of Wisconsin - Madison, Madison, WI, USA. [5]UW Carbone Cancer Center, School of Medicine and Public Health, University of Wisconsin - Madison, Madison, WI, USA. ✉e-mail: junsu.kang@wisc.edu

regeneration and fibrosis is vital to understand the intricate processes of heart regeneration.

Here, we show regenerative and fibrotic effects of *interleukin 11a* (*il11a*) in the zebrafish heart. Since these two outcomes of *il11a* are uncoupled, we devise a novel approach to leverage the regenerative function of *il11a* by inhibiting ERK activity, which is a specific mediator of *il11a* for fibrosis. Our findings propose a novel therapeutic strategy with *Il11* signaling for cardiac repair by governing regeneration and fibrosis.

## Results

### Dynamics of endogenous *il11a* expression during zebrafish heart regeneration

Interleukin 11 (IL11), a pleiotropic cytokine of the IL6 family, plays regenerative roles in multiple injury contexts[18–20]. To investigate roles of *il11a*, one of two *il11* zebrafish homologs, in zebrafish hearts, we analyzed its expression with a newly created knock-in reporter line, *il11a^EGFP* (Fig. 1a, b and Supplementary Fig. 1a). *il11a^EGFP* expression is undetectable in uninjured hearts of embryonic and adult hearts (Fig. 1a, b and Supplementary Fig. 1b, c), suggesting its minimal contribution to heart development and homeostasis. Upon cardiac injury, *il11a* is robustly induced from non-CMs, primarily epicardium and endocardium (Fig. 1a–d and Supplementary Fig. 1b, c). *il11a^EGFP* expression was detectable as early as 1 dpa, highly and widely induced at 3 dpa throughout the ventricle, became localized to the wound site at 7 dpa, and largely undetectable by 14 dpa (Fig. 1a, b). Additionally, our further analysis to dissect *il11a* expressing cells revealed that epicardial *il11a^EGFP* expression decreased, whereas endocardial expression at the wound area remained constant from 3 to 7 dpa (Supplementary Fig. 1d, e). The robust and prompt *il11a^EGFP* expression throughout the

ventricle during the early phase of regeneration suggests its potential influence on the activation of non-CMs, whereas wound-restricted *il11a^EGFP* expression during the mid-phase of regeneration indicates a potential interaction between *il11a* signaling and CMs for their renewal. The diminished expression of *il11a* by 14 dpa indicates the minimal contribution of *il11a* during the late phase of regeneration (Supplementary Fig. 1f, g).

### *il11a* induction can promote cardiomyocyte proliferation in uninjured hearts

Previous studies of *il11* signaling in zebrafish hearts mainly focused on loss-of-function experiments with mutants of *il11* signaling components, such as *il11ra*, revealing impaired regeneration in the mutant hearts[21]. In rodents, *Il11* roles on cardiac fibrosis have been reported, but its regenerative roles remain unclear[22–24]. To our knowledge, there has been no comprehensive analysis of *il11* functionality on cell cycle reentry of CMs. To address this, we developed a new transgenic line, *Tg(actb2:loxp-BFP-STOP-loxp-il11a)*, or *actb2:BS-il11a*, which allowed conditional overexpression (OE) of *il11a* in combination with *CreER* line. Crossing with *cmlc2:CreER*, we induced *il11a*OE from cardiac muscle cells upon tamoxifen treatment in the absence of injury (Fig. 2a). Notably, *il11a*OE led to the marked induction of cell cycle reentry in CM of the uninjured hearts, whereas control hearts showed little to no cell cycle activity in CMs (Fig. 2b, d). Moreover, we noted a significant induction of phospho-H3 (pH3) positive CMs in the uninjured *il11a*OE hearts, compared to control (Supplementary Fig. 2a, b), suggesting that *il11a*OE is sufficient to trigger CM proliferation. *il11a*-triggered CM proliferation could be caused by cell death or damages. To test this possibility, we performed Terminal deoxynucleotidyl transferase (TdT) dUTP Nick-End Labeling (TUNEL) assay and found

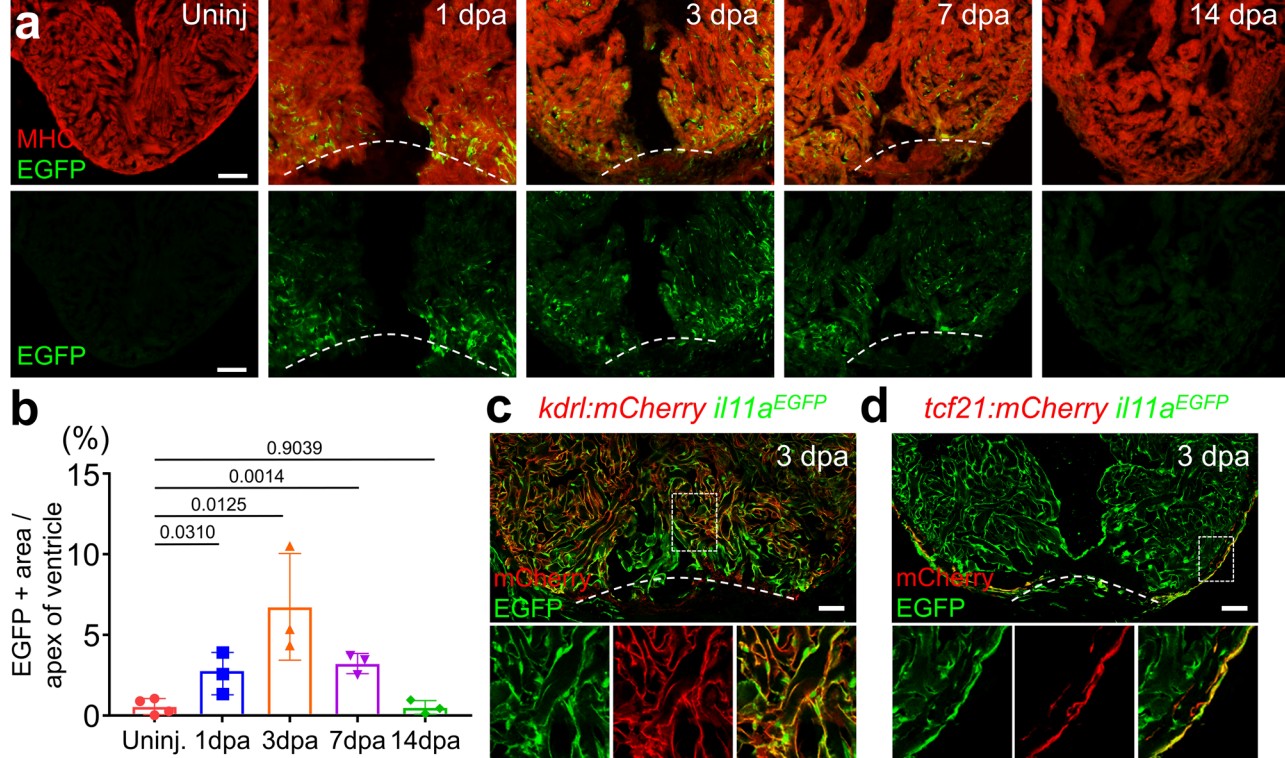

**Fig. 1 | Spatiotemporal *il11a* expression during heart regeneration in adult zebrafish. a** Representative cardiac section images of *il11a^EGFP*. Uninj, Uninjured. dpa, days post-amputation. **b** Quantification of EGFP expression area near the wound area. Biological replicates = 4, 3, 3, and 3 for uninjured and 3, 7, and 14 dpa hearts, respectively. Comparison of *il11a^EGFP* expressing cells with *kdrl:mCherry* (**c**) and *tcf21:mCherry* (**d**) in 3 dpa hearts. *kdrl:mCherry* and *tcf21:mCherry* represent endocardium and epicardium, respectively. Dash boxes correspond to the region magnified in the panels below. Dotted lines demarcate the amputation plane. Scale bars, 100 μm in (**a**), 50 μm in (**c**) and (**d**). Data are mean ± SEM in (**b**). *p*-values were determined by one-way ANOVA with Dunnett's multiple comparison test in (**b**).

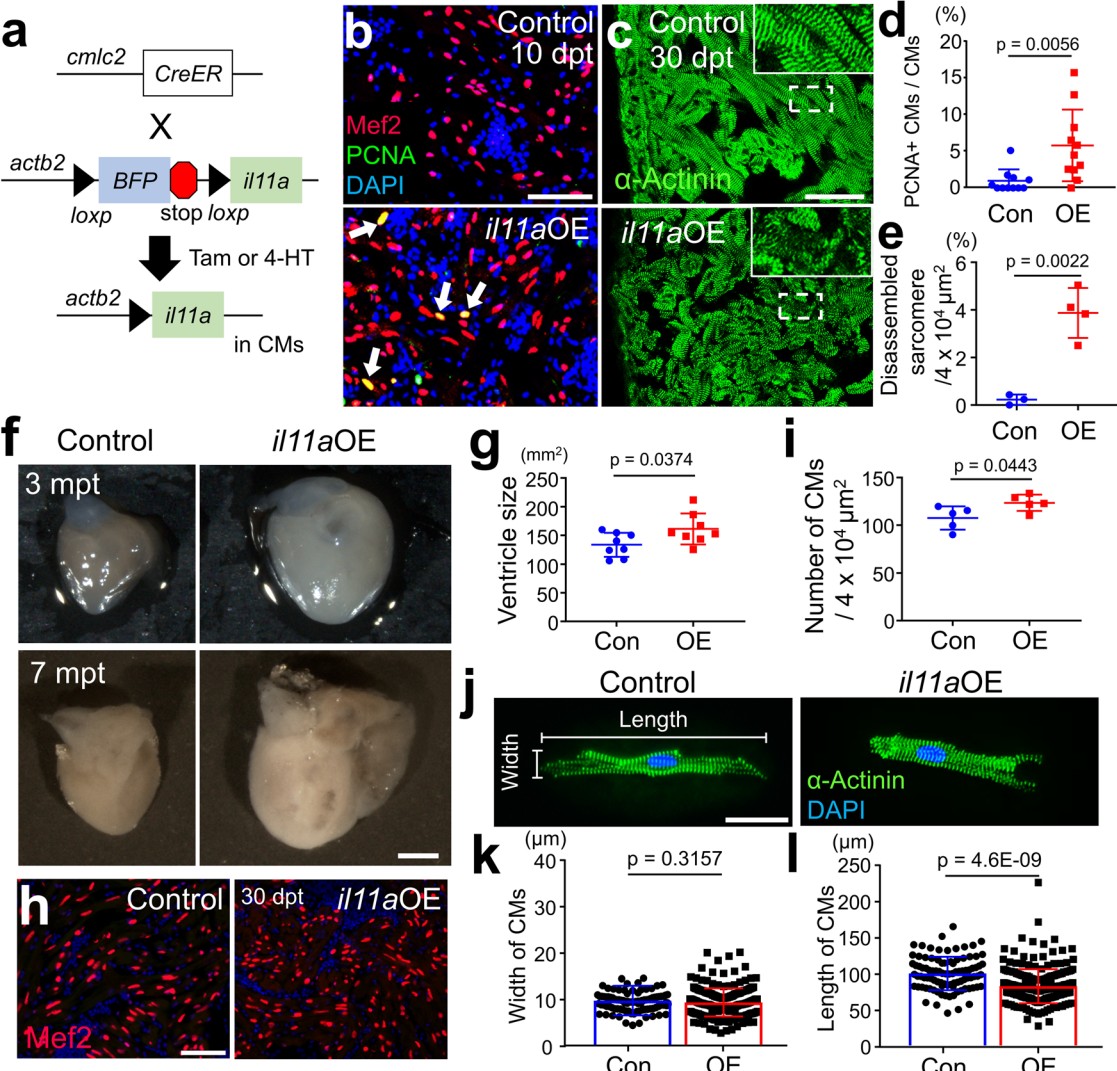

**Fig. 2 | Myocardial *il11a* overexpression (OE) triggers CM proliferation in the absence of injury. a** Schematic of tamoxifen (Tam) or 4-HT (4-Hydroxytamoxifen) inducible *il11a* overexpression (OE). *Tg(actb2:loxp-TagBFP-STOP-Loxp-il11a)*; *cmlc2:CreER* were administered with Tam or 4-HT to overexpress *il11a* (*il11a*OE). *CreER*-negative littermates were used as controls. **b** Representative cardiac section images stained with Mef2 (Red) and PCNA (Green) from control and *il11a*OE at 10 days post-treatment (dpt). Arrows indicate proliferative CMs. **c** Representative cardiac sections images stained with α-Actinin (green) from control and *il11a*OE uninjured hearts at 30 dpt. Insets correspond to higher magnifications of dashed boxes. **d** The percentage of PCNA+ CMs of uninjured hearts from control (Con) and *il11a*OE (OE) at 7 dpt. Biological replicates = 11 for control and *il11a*OE. **e** Quantification of disorganized sarcomere structures in the ventricles of Con and *il11a*OE at 30 dpt. Biological replicates = 3 and 4 for control and *il11a*OE, respectively. **f** Whole-mount images of 3- and 7-months post treatment (mpt) hearts from Con and *il11a*OE. **g** Quantification of the ventricle size from 3 mpt Con and *il11a*OE. Biological replicates = 8 for control and *il11a*OE. **h** Representative cardiac section images of 30 dpt control and *il11a*OE stained with Mef2 (red) and DAPI (blue). **i** Quantification for CM numbers in the ventricles from 30 dpt Con and *il11a*OE. Biological replicates = 5 for control and *il11a*OE. **j** Representative images of dissociated ventricular CMs stained with α-Actinin (green) from 2 mpt control and *il11a*OE. Quantification of the width (**k**) and the length (**l**) of individual CMs. Biological replicates = 92 and 234 for control and *il11a*OE, respectively. Scale bars, 50 μm in (**b**), (**c**) and (**h**), 0.5 mm in (**f**), 25 μm in (**j**). Data are mean ± SEM in (**d**), (**e**), (**g**), (**i**), (**k**), and (**l**). *p*-values were determined by unpaired two-tailed *t*-test in (**d**), (**e**), (**g**), (**i**), (**k**), and (**l**).

---

that TUNEL+ cells are undetectable in *il11a*OE hearts, while our control experiment with cryoinjured hearts displayed robust TUNEL+ cells at 7 days post cryoinjury (dpi) (Supplementary Fig. 3). Our results indicate that *il11a*OE unlikely induces apoptosis but directly triggers CM proliferation.

We next analyzed bulk RNA-seq profiles with control and *il11a*OE ventricles at 7 days post treatment (dpt), identifying 1327 genes (n = 862 and 465 for up- and downregulated genes, respectively) with significantly changed expression levels (adjusted *P* [*P*-adj] <0.05; fold change > 4) (Supplementary Fig. 2c and Supplementary Data 1). *il11a*OE upregulated regeneration-associated genes, including a regeneration-specific factor (*lepb*), pro-regenerative ECM proteins

(*fn1b, hmmr*, and *col12a1b*), and cell cycle genes (*plk1, cdk1, ccnb1, ccnb2, and mki67*), indicating activated regeneration response by *il11a* induction[3,25–28]. Gene ontology (GO) analyses with upregulated genes in *il11a*OE hearts revealed enrichment in cell cycle, nuclear division, and DNA replication, highlighting active proliferation (Supplementary Fig. 2d). Additionally, gene set enrichment analysis (GSEA) identified significant enrichment of cell cycle signature in *il11a*OE (Supplementary Fig. 2e). *il11a* induction also highly upregulated key components of janus kinase/signal transducers and activators of transcription (JAK/STAT) signaling pathway, including *jak1, stat3*, and *socs3b*, which constitute a canonical downstream pathway of IL11[29–31] (Supplementary Fig. 4a). Our in situ hybridization (ISH) analysis further confirmed

that JAK/STAT-related factors are highly upregulated in CMs upon *il11a* induction (Supplementary Fig. 4b). Moreover, *il11a*OE robustly increased phospho-STAT3 (p-STAT3) labeled CMs and 5.87% of pSTAT-3[+] Mef2[+] CMs are labeled by EdU, suggesting the active STAT3 pathway in CMs (Supplementary Fig. 4c, d).

To re-enter cell cycle, CMs undergo dedifferentiation, characterized by sarcomere disassembly and downregulation of genes associated with mature CMs[4,32–34]. Indeed, the downregulated genes in response to *il11a* induction are primarily associated with cardiac muscle contraction and metabolic processes (Supplementary Fig. 2d, f), suggesting a transcriptional shift from a mature to an immature state in CMs upon *il11a* induction. We next assessed the impact on sarcomere structure through α-Actinin staining in control and *il11a*OE at 30 dpt. While control hearts displayed no sign of dedifferentiated CMs, *il11a*OE hearts exhibited a markedly increased sarcomere disassembly with blurred Z-discs (Fig. 2c, e). To evaluate the long-term mitogenic effect of *il11a*OE, we analyzed the uninjured hearts at 3 and 7 months post tamoxifen treatment (mpt). While no differences in body size and weight were observed between control and *il11a*OE (Supplementary Fig. 5), *il11a*OE hearts displayed a noticeable enlargement with a 20% increase in size at 3 mpt (Fig. 2f, g) and the significantly augmented size of the ventricles at 7 mpt (Fig. 2f), indicative of cardiomegaly.

Enlarged hearts can be caused by cardiac hypertrophy, a prominent mechanism responsible for the augmented mammalian hearts following injury[35], or hyperplasia, a desirable regenerative feature. To determine hypertrophic or hyperplastic effect of *il11a* induction, we quantified the CM density in the ventricles. The number of Mef2[+] nuclei in the ventricle significantly increased in *il11a*OE, compared to control (123.45 ± 8.71 and 107.45 ± 12.23 CMs/4 × 10^4 μm^2 for *il11a*OE and control, respectively) (Fig. 2h, i). Additionally, the individual CMs isolated from *il11a*OE hearts exhibited smaller length (83.65 ± 23.5 μm) than control (101.2 ± 23.2 μm) (Fig. 2j, l). While there is no notable difference in width between control and *il11a*OE CMs, *il11a*OE hearts likely contain some enlarged CMs, indicating a possible low level of hypertrophic effect (Fig. 2j, k). Collectively, these results indicate that *il11a* activation can lead to notable myocardial hyperplasia, highlighting its regenerative effect in zebrafish hearts.

### *il11a* overexpression hypervascularizes the uninjured zebrafish hearts

Stimulating revascularization of coronary vessels is a crucial regenerative process for the hearts[36–38]. To determine whether *il11a*OE can promote coronary vascularization in the absence of injury, we examined the coronary network using *fli1a:EGFP*[+], which labels endothelial vessels[39,40]. Remarkably, whole-mount images demonstrated a more elaborate *fli1a:EGFP*[+] coronary network in the *il11a*OE hearts as significantly increased vessel density was observed in *il11a*OE ventricles (18.84 ± 4.98% to 26.34 ± 3.27%) (Fig. 3a, b). *fli1a:EGFP*[+] area in the cortical layer is also highly enhanced in *il11a*OE hearts (10.58 ± 1.19%), compared to control (6.78 ± 0.8%) (Fig. 3c, d). Our transcriptome data revealed that *il11a*OE hearts exhibit a significant upregulation of genes associated with vascularization, including *angptl2a, emilin2a, hmmr, rspo3, sulf1*, and *thsd7aa*[37,41–45] (Supplementary Fig. 6a). Our scRNA-seq analysis with published dataset of regenerating hearts[44] identified their injury-dependent induction predominantly within the epicardial cell clusters (Fig. 3e and Supplementary Fig. 6b–d). Our ISH analysis also validated that the angiogenic factors are highly upregulated in the epicardial and cortical layer in *il11a*OE, but not in control (Fig. 3f). Given the pivotal role of the epicardium in guiding coronary growth[8,37,41,46], alongside with angiogenic factor increment by *il11a*, our data suggest that *il11a* induction activates epicardium to facilitate coronary growth.

While epicardium is normally quiescent in uninjured hearts, cardiac injury activates epicardium to undergo epithelial-to-mesenchymal transition (EMT)[47,48], giving rise to epicardial progenitor cells (EPCs), which are marked by *col12a1b* expression[49,50]. These EPCs subsequently differentiate into diverse epicardium-derived cells (EPDCs) in regenerating hearts[49]. We thus examined whether *il11a* induction can activate the dormant epicardium to initiate the generation of *col12a1b*[+] EPCs even in the uninjured hearts. In control hearts, *tcf21*[+] epithelial cells are typically restricted to one or two layers outlining the ventricles. In contrast, within *il11a*OE hearts, *tcf21*[+] layers expanded into the mesenchymal layers, a deeper region of the cortical wall, indicating that *il11a* induction triggers the epicardium to undergo EMT (Fig. 3g). Evidently, a subset of *tcf21*[+] epicardial cells within *il11a*OE hearts display *col12a1b:EGFP* expression at 10 dpt, indicating that *il11a* induction activates *tcf21*[+] epicardial cells to become EPCs (Fig. 3g). By 30 dpt, *il11a*OE hearts displayed extensive expansion of *col12a1b*[+] EPC populations within the deeper mesenchymal layer (Fig. 3g). In contrast, *col12a1b*[+] EPCs were conspicuously absent in control hearts at 10 and 30 dpt. We also tested whether *il11a*OE can stimulate the activation of *hapln1b*[+] EPDCs as *hapln1*[+] EPDCs are known to impact coronary growth[8]. Our data revealed that *il11a*OE hearts display the expansion of *hapln1b*[+] cell populations in the mesenchymal layer at 30 dpt (Fig. 3h, i). Therefore, our results indicate that *il11a* can stimulate the epicardium to generate EPCs which subsequently differentiate into EPDCs impacting coronary vascularization.

### The cardiac fibrotic effects of prolonged *il11a* treatment

Given recent reports indicating IL11 as a fibrotic factor[22–24], we explored whether *il11a* could exert a fibrotic role in zebrafish hearts. At 30 dpt, there is no significant difference of collagen deposition between control and *il11a*OE hearts (Fig. 4a). While fibrotic tissue was undetectable in control hearts, *il11a*OE hearts exhibited collagen accumulation in the cortical layer by 3 mpt and massive collagen deposition by 7 mpt (Fig. 4a, b). The cardiac fibrosis mainly manifested in the cortical layer and surface, rather than within the ventricles, and its thickness expanded over time (Fig. 4a, b). These findings suggest that long-term *il11a* induction leads to cardiac fibrosis, especially in the cortical layer.

To assess the induction of fibrosis-associated genes by prolonged *il11a*OE, we performed RNA-seq analysis with 7 mpt control and *il11a*OE hearts, comparing them with 7 dpt transcriptome datasets. The border zone of the injured hearts exhibits high enrichment of embryonic CM genes, including *nppa, nppb, desma*, and *ankrd1a*[51–53]. These embryonic genes, in addition to cell cycle and regenerative ECM genes (Supplementary Figs. 2c and 7a), are upregulated early at 7 dpt after *il11a* induction, suggesting the prompt regenerative effect of *il11a* in hearts (Supplementary Fig. 7a). By contrast, our transcriptome analysis revealed a significant upregulation of pathological fibrotic and cardiac fibroblast marker genes, including *col1a1a/b, col5a1, postnb, sox9a/b, dcn*, fibroblast activation genes (*cthrc1a, thbs4b*), ECM-linking genes (various Lysyl oxidase (LOX) enzymes), ECM maturation genes (*mgp, sparc*) and in 7 mpt *il11a*OE hearts, but not 7 dpt (Fig. 4c, Supplementary Fig. 7a, b, and Supplementary Data 1 and 2)[13,54–62,59–62]. Extended *il11a* induction moderately, but not robustly, upregulates one of two *transforming growth factor β1* (*tgfb1*) homologs (Supplementary Fig. 7a), aligning with the previous finding that *Il11* plays roles as a downstream effector of *Tgfb1* for cardiac fibrosis[22]. These results highlight the dual functions of *il11a* as a regenerative factor with more immediate effects and a fibrotic factor, presumably, with indirect consequences.

In regenerating zebrafish hearts, cardiac fibrosis is largely mediated by epicardium-derived fibroblasts that appear in the cortical layer following injury[6,13,63,64]. Given that the fibrosis driven by *il11a* overexpressing hearts is predominantly observed in this cortical layer, we hypothesize that prolonged *il11a* exposure can activate the epicardium constantly, leading to sustained cardiac fibroblast production and subsequent fibrosis. Supporting this notion, while Periostin[+] (Postn[+]) cells are undetectable in the 10 dpt *il11a*OE hearts, *il11a* induction led

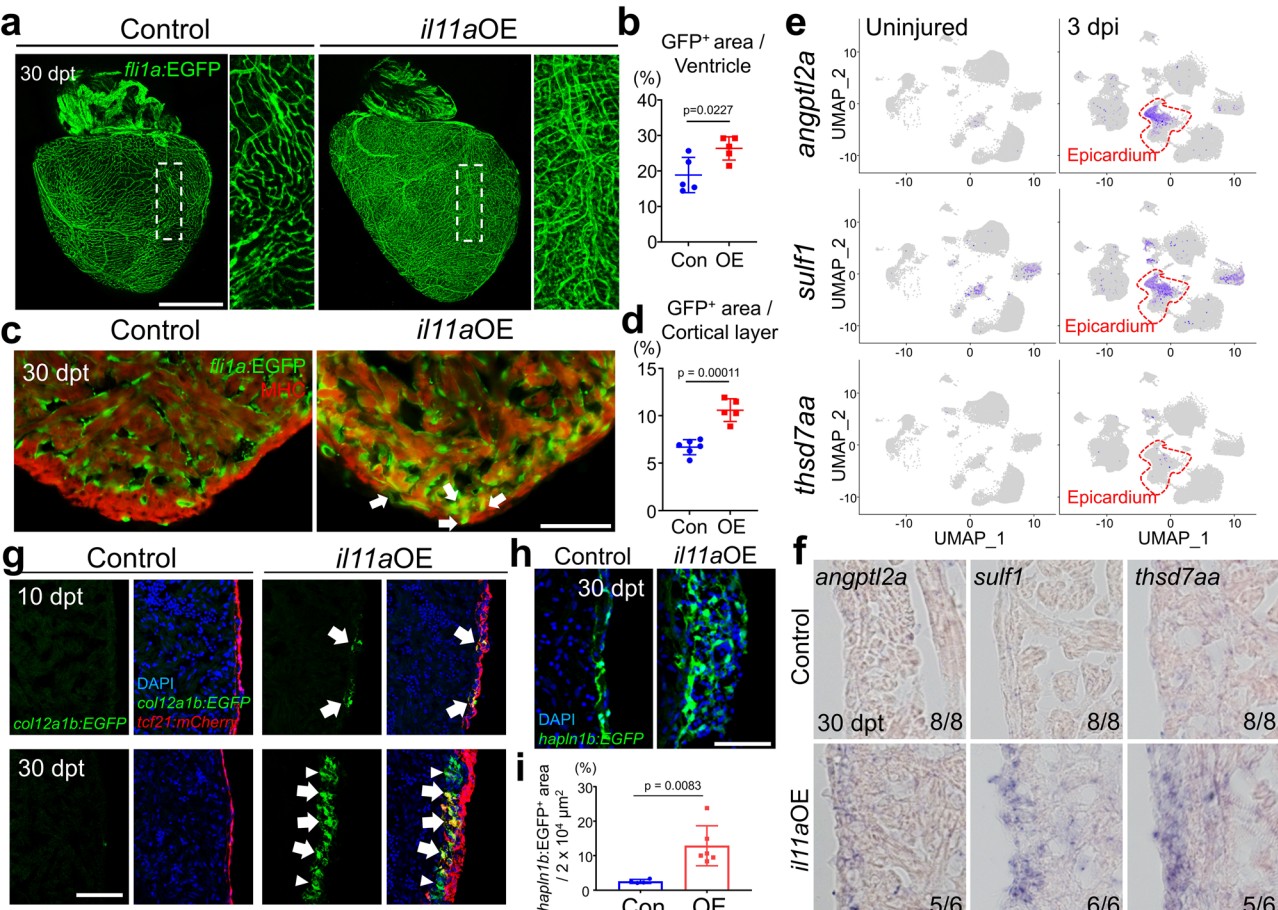

**Fig. 3 | *il11a* induction stimulates vascularization in the absence of injury.**
**a** Representative whole-mount images of 30 dpt *fli1a:EGFP* hearts from control and *il11a*OE. Insets correspond to higher magnifications of dashed boxes.
**b** Quantification of *fli1a:EGFP*⁺ vessel area in the ventricles for control Con and *il11a*OE. Biological replicates = 5 for control and *il11a*OE. **c** Representative cardiac section images of 30 dpt control and *il11a*OE hearts. Green and red indicate *fli1a*:EGFP and myosin heavy chain (MHC), respectively. Arrows indicate expanded EGFP⁺ region in the cortical layer. **d** Quantification of EGFP⁺ area in the cortical muscle layer from Con and *il11a*OE. Biological replicates = 6 and 5 for control and *il11a*OE, respectively. **e** Gene expression plots of *angptl2a, sulf1*, and *thsd7aa* induced in epicardium at 3 days post injury (dpi). Cells expressing the indicated gene are colored purple, and the relative intensity indicates relative expression levels. Red dashed lines indicate epicardial expression of indicated genes. uniform manifold approximation and projection (UMAP) clustering with annotated cell-types are in Supplementary Fig. 6. **f** Representative images of in situ hybridization (ISH) for angiogenic factors (*angptl2a, sulf1, thsd7aa*) on cardiac sections of control and *il11a*OE. The number in the lower right corner of each image represents the fraction of the analyzed hearts with displayed phenotype. Biological replicates = 6 and 8 for control and *il11a*OE, respectively. **g** Representative cardiac section images of control and *il11a*OE expressing *tcf21:mCherry; col12a1b:EGFP*. Top, 10 dpt. Bottom, 30 dpt. Blue, DAPI. Green, EGFP. Red, mCherry. Arrows indicate *tcf21:mCherry* and *col12a1b:EGFP* double positive cells. Arrowheads indicate *tcf21:mCherry; col12a1b:EGFP*⁺ cells. **h** Representative cardiac section images of 30 dpt control and *il11a*OE hearts. Green indicates *hapln1b*:EGFP. **i** Quantification of *hapln1b*:EGFP⁺ area in the cortical muscle layer from Con and *il11a*OE. Biological replicates = 4 and 6 for control and *il11a*OE, respectively. Scale bars, 500 μm in (**a**), 50 μm in (**c**), (**f**), (**g**), and (**h**). Data are mean ± SEM in (**b**), (**d**), and (**i**). *p*-values were determined by unpaired two-tailed *t*-test in (**b**), (**d**), and (**i**).

to the emergence of Postn⁺ fibroblasts in the cortical layer at 30 dpt without an injury (Fig. 4d, e). These Postn⁺ fibroblasts are gradually expanded at 3 mpt and dispersed across the cortical layer, where collagens are extensively deposited. Next, we examined whether epicardial cells can give rise to the Postn⁺ fibroblasts. Notably, 66% Postn⁺ cells are colocalized with *tcf21*:mCherry⁺ epicardial cells in *il11a*OE at 30 dpt while there is no Postn expression in epicardial cells of control hearts (Fig. 4f, g). Thus, our results indicate that fibrosis resulting from ectopic *il11a* expression is mediated by epicardial-derived Postn⁺ cardiac fibroblasts.

ACTA2, smooth muscle alpha (α)−2 actin, is used as a biomarker for the injury-induced myofibroblasts in mammalian hearts[65,66], raising a question of ACTA2 expression in response to *il11a*OE. In addition to myofibroblasts, ACTA2 also labels vascular smooth muscle cells (VSMCs) and dedifferentiating CMs[51,67]. While circular ACTA2⁺ cell layers were observed in the cortical layers of both control and *il11a*OE

hearts, *il11a* induced the additional ACTA2⁺ cells in the cortical layer without an injury at 30 dpt (Supplementary Fig. 8a–e). To discriminate ACTA2-expressing cells, we examined their location and colocalization with blood vessels and CM markers. In control hearts, most ACTA2⁺ cells were found encircling *fli1a:EGFP*⁺ coronary vessels within the outer cortical layer of ventricles, an indicative of VSMC (Supplementary Fig. 8c). Although circular ACTA2⁺ VSMC layers were also evident in *il11a*OE hearts, we found that *il11a*OE hearts additionally display a thick and linear ACTA2⁺ layer in the deeper cortical layer (Supplementary Fig. 8c). These ACTA2⁺ layers are not closely associated with *fli1a:EGFP*⁺ cells, indicating non-VSMC cell types (Supplementary Fig. 8d). While the majority of these ACTA2⁺ cells lack α-Actinin CM signature (Supplementary Fig. 8e), they co-express a myosin heavy chain (MHC) CM gene (Supplementary Fig. 8f–h). Based on the anatomical location of the primordial CM layer that gives rise to trabecular CMs during development[68], our results indicate that *il11a*-induced

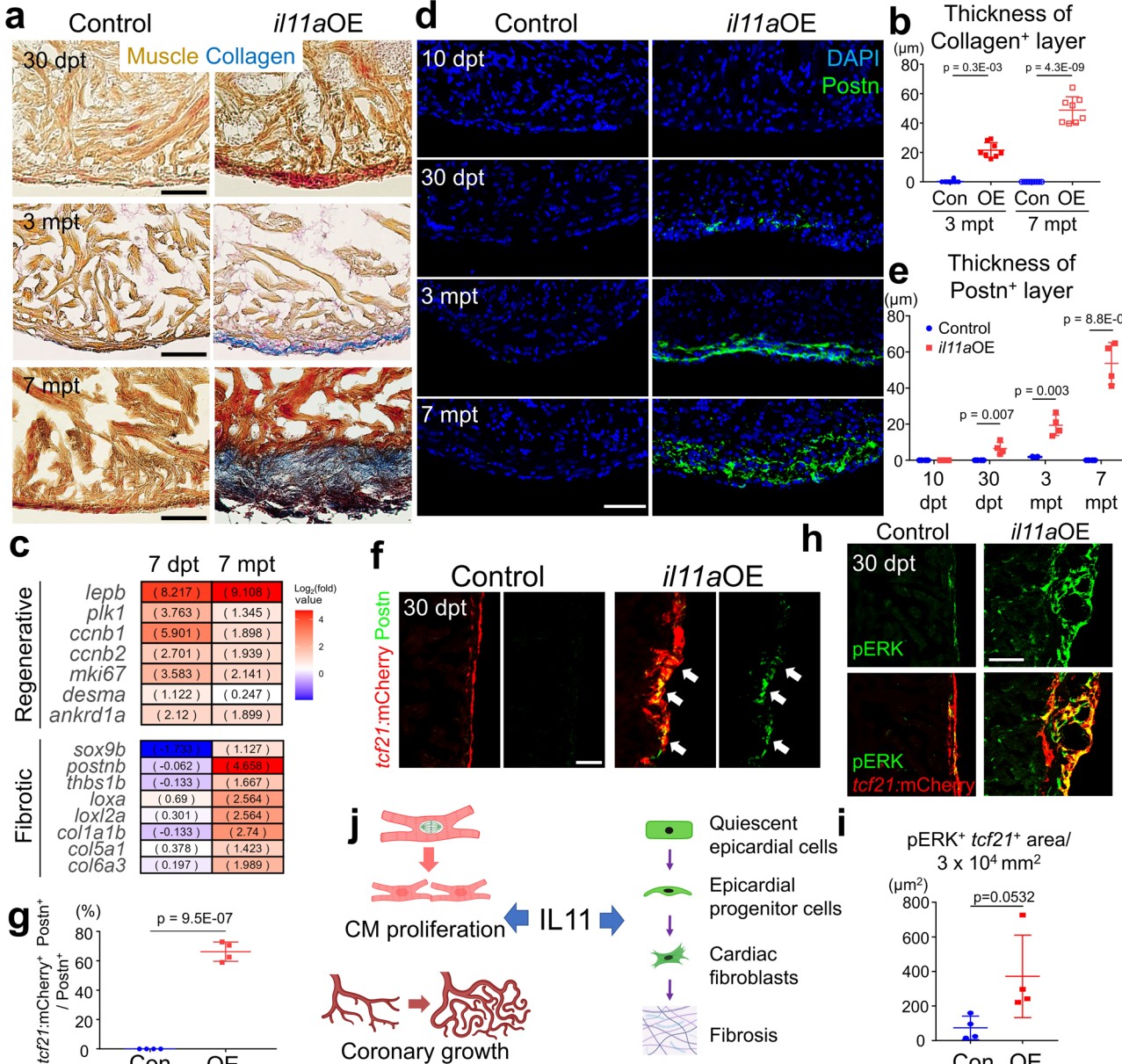

**Fig. 4 | Fibrotic roles of *il11a* in adult zebrafish hearts. a** Representative AFOG staining images of uninjured control and *il11a*OE hearts at 30 dpt (top), 3 mpt (middle), and 7 mpt (bottom). **b** Thickness of the collagen⁺ layer from Con and *il11a*OE at 3 and 7 mpt. Biological replicates = 8, 9, 9, and 8 for 3 mpt control, 3 mpt *il11a*OE, 7 mpt control, and 7 mpt *il11a*OE, respectively. **c** Differential expression of genes associated with regeneration (top), cardiac fibroblasts and pathological fibrosis (bottom) between 7 dpt and 7 mpt. Gene associated with fibrosis and fibroblasts are significantly upregulated in 7 mpt *il11a*OE hearts, compared to control, while those genes are not induced in 7 dpt *il11a*OE hearts. Expanded list of genes is in Supplementary Fig. 7a. **d** Representative cardiac section images of control and *il11a*OE hearts. Green and blue indicate Periostin (Postn)⁺ fibroblasts and DAPI, respectively. **e** Thickness of the Postn⁺ layer from Con and *il11a*OE at 10 and 30 dpt and 3 and 7 mpt. Biological replicates = 4 for all conditions, except 3 mpt control (*n* = 3). **f** Representative images of cardiac sections labeled with Postn

(green) and *tcf21*:mCherry (red) at 30 dpt. Arrows indicate *tcf21*⁺ epicardial cells expressing Postn. **g** Colocalization assay of Postn⁺ cells with *tcf21*:mCherry⁺ layer. Biological replicates = 4 for control and *il11a*OE. **h** Representative cardiac section images stained with mCherry (red) and pERK (green) of 30 dpt Con and *il11a*OE hearts expressing *tcf21:mCherry*. **i** Quantification of pERK⁺ area in *tcf21*⁺ epicardium in the ventricles for Con and *il11a*OE. Biological replicates = 4 for control and *il11a*OE. **j** Dual roles of IL11 in the hearts. IL11 triggers cardiac regeneration programs, including CM proliferation and revascularization. Simultaneously, IL11 contributes to fibrosis by stimulating quiescent epicardial cells to generate epicardial progenitor cells (EPCs), which give rise to cardiac fibroblasts. Created in BioRender. Kang, J. (2024) BioRender.com/g25e465. Scale bars, 25 μm (top and middle) and 50 μm (bottom) in (**a**), 50 μm in (**d**), (**f**), and (**h**). Data are mean ± SEM in (**b**), (**e**), (**g**) and (**i**). *p*-values were determined by two-stage step-up multiple *t*-test in (**b**) and (**e**) and by unpaired two-tailed *t*-test in (**g**) and (**i**).

ACTA2⁺ layers in the deeper cortical layer are likely dedifferentiating CMs.

As ERK, a downstream effector of the MAPK pathway, controls various cellular processes, including proliferation, survival, and fibrosis[69,70], we questioned whether ERK mediates *il11a*-inducible fibrosis in the epicardium. While control hearts contain a minimal

number of phospho-ERK (pERK)⁺ *tcf21*⁺ epicardial cells, their numbers are significantly increased upon *il11a* induction (Fig. 4h, i and Supplementary Fig. 7c, d). Also, the pERK signal remained undetectable in CMs in both control and *il11a*OE (Supplementary Fig. 7e), indicating the limited impact of MAPK/ERK signaling on CM proliferation. These findings suggest that the persistent *il11a* induction can provoke

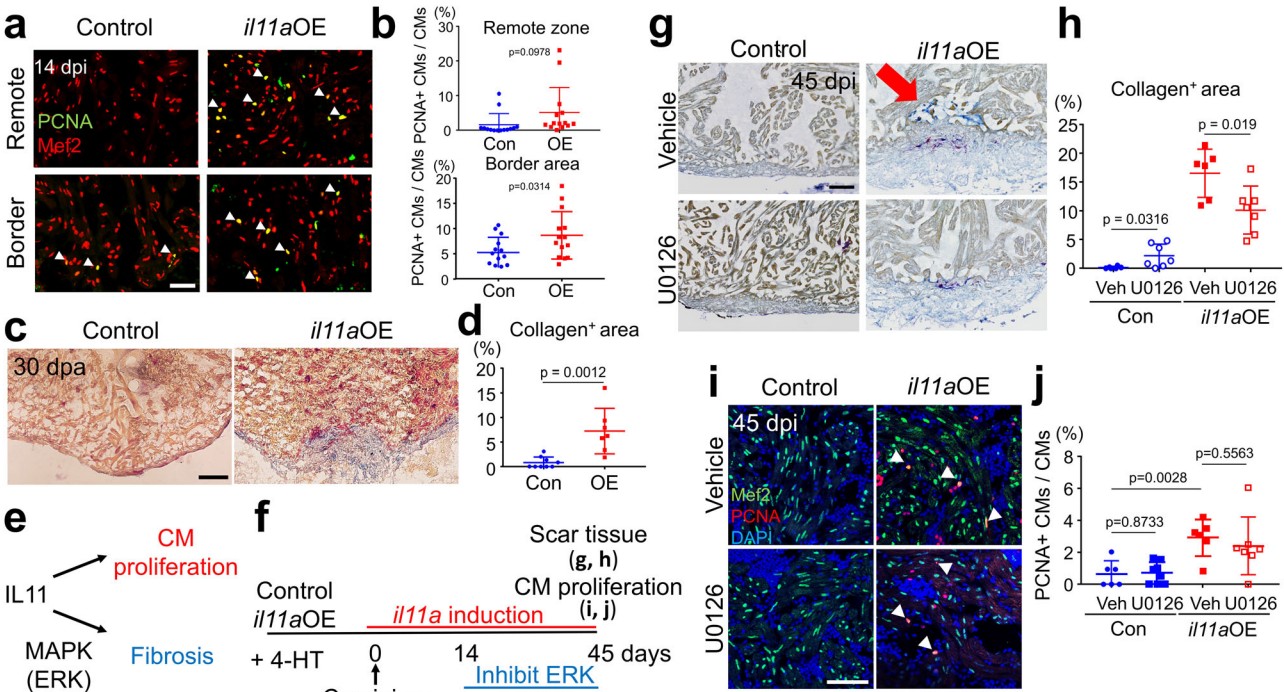

**Fig. 5 | Combinatorial treatment enables to harness the regenerative potential of *il11a* in injured hearts. a** Representative cardiac section images of 14 dpi control and *il11a*OE hearts stained with Mef2 (red) and PCNA (green). Cryoinjury was performed one day after administering four daily injections of 4-HT. Arrowheads indicate proliferative CMs. **b** The percentage of PCNA⁺ CMs in the remote and border zone from Con and *il11a*OE at 14 dpi. Biological replicates = 14 for control and *il11a*OE. **c** Representative AFOG staining images of control and *il11a*OE hearts at 30 dpa. **d** Quantification of collagen⁺ area in the ventricle of control Con and *il11a*OE at 30 dpa. Biological replicates = 7 and 9 for control and *il11a*OE, respectively. **e** Dual roles of *il11a* in injured hearts. *il11a* promotes CM proliferation while inducing fibrosis via Extracellular signal-regulated kinase (ERK) activity. **f** Experimental scheme to harness the regenerative potential of *il11a* while mitigating fibrosis by treating U0126, an ERK inhibitor, starting at 14 dpi. **g** Representative AFOG staining images of vehicle (Veh, 0.1% DMSO) or U0126

(10 μM) treated hearts from control and *il11a*OE. Hearts were collected at 45 dpi/55 dpt. The arrow indicates the presence of scar tissue near the injury site in Veh-treated *il11a*OE hearts. **h** Quantification of scar area in the ventricles of Veh or U0126-treated control and *il11a*OE hearts. Biological replicates = 6, 7, 6, and 7 for control_DMSO, control_U0126, *il11a*OE_DMSO, and *il11a*OE_U0126, respectively. **i** Representative cardiac section images of Veh- or U0126-treated control and *il11a*OE hearts stained with Mef2 (green) and PCNA (red) at 45 dpi. Arrowheads indicate proliferative CMs. **j** The percentage of PCNA⁺ CMs in the ventricles of Veh or U0126-treated control and *il11a*OE hearts at 45 dpi. Biological replicates = 6, 7, 6, and 7 for control_DMSO, control_U0126, *il11a*OE_DMSO, and *il11a*OE_U0126, respectively. Scale bars, 25 μm in (**a**) and (**g**), 50 μm in (**c**) and (**i**). Data are mean ± SEM in (**b**), (**d**), (**h**) and (**j**). *p*-values were determined by unpaired two-tailed *t*-test in (**b**), (**d**), (**h**) and (**j**).

fibrosis by promoting the presence of cardiac fibroblasts from MAPK/ERK-activated EPCs.

## *il11a* induction with ERK inhibition can enhance CM proliferation without exacerbating fibrosis

Our studies on *il11a*OE in uninjured hearts demonstrated dual roles of *il11a*, acting as a double-edged sword in heart repair: *il11* signaling can promote the regenerative program while also contributing to cardiac fibrosis (Fig. 4j). We next explored whether this dual functionality of *il11a* is also operational in injured hearts. At 14 days post injured hearts, ectopic *il11a* activation significantly increased CM cycling (8.61 ± 1.26% and 5.20 ± 0.80% in OE and control, respectively) (Fig. 5a, b and Supplementary Fig. 9a), confirming its regenerative effect on injured hearts. To assess the fibrotic effects of *il11a* on injured hearts, we performed ventricular apex resection with control and *il11a*OE fish and collected hearts at 30 dpa, at which myocardial wall regeneration is typically completed[1]. Control hearts had limited scarring at the wound, a sign of successful scar resolution. In contrast, *il11a*OE hearts exhibited apparent collagen-rich scar tissues at the injury site (8.19 ± 3.33% and 0.62 ± 0.67% in OE and control, respectively) (Fig. 5c, d and Supplementary Fig. 9b, c). This time point is relatively earlier than when *il11a*-stimulated fibrosis is observed in the uninjured hearts, as uninjured *il11a*OE hearts did not display noticeable collagen-rich fibrotic tissues at 30 dpt (Fig. 4a). Furthermore, ectopic *il11a* induction led to

the persistence of *col12a1b*⁺ EPCs and fibrotic tissues at the injury site in cryoinjured hearts (Supplementary Fig. 10a–d). Similar to *il11a*OE uninjured hearts, we observed prominent Postn⁺ layers at the wound area of *il11a*OE hearts, suggesting the existence of cardiac fibroblasts induced by *il11a*OE (Supplementary Fig. 10e, f). Overall, these results indicate regenerative and fibrotic functions of *il11a* in injured hearts.

To harness the regenerative potential of *il11a* for injured hearts, mitigating its fibrotic function is crucial. Thus, we devised a concurrent treatment strategy involving *il11a* induction with preventing cardiac fibrosis through ERK/MAPK inhibition (Fig. 5e). This approach aims to facilitate heart repair by promoting CM proliferation via *il11a* induction while minimizing cardiac fibrosis by inhibiting fibrosis-specific signaling at a later time point. To achieve this, we administered U0126, an ERK inhibitor[71], for a month, starting at 2 weeks after cryoinjury with control and *il11a*OE fish and collected hearts at 45 days post injury (dpi) (Fig. 5f). Collagen deposition was not evident in the hearts of control fish treated with vehicle or U0126 (Fig. 5g, h), suggesting that U0126 treatment during the late regenerative phase minimally affects heart repair. By contrast, *il11a*OE hearts treated with vehicle displayed significant collagen deposition at the injury site, confirming the fibrotic function of *il11a* in injured hearts (Fig. 5g, h). Interestingly, U0126 treatment in *il11a*OE significantly reduced the presence of fibrotic tissues at the injury site (Fig. 5g, h), implicating that inhibiting ERK activity reduces *il11a*-induced cardiac fibrosis. Moreover, we observed

that CMs in *il11a*OE hearts treated with U0126 and vehicle exhibited a similar cell cycling rate (Fig. 5i, j). As another readout of regeneration, we quantified the total number of Mef2[+] CMs throughout a ventricle in 45 dpi hearts and found that U0126-treated *il11a*OE hearts have significantly more Mef2[+] CMs than control hearts (Supplementary Fig. 11a, b). These results highlight the specific effect of ERK/MAPK inhibition on cardiac fibrosis with limited influence on regeneration. Taken together, our findings provide a proof-of-principle that combinatorial treatment of *il11a* induction with inhibiting fibrosis holds significant potential for enhancing heart regeneration following cardiac injury.

## Discussion

Our expression analysis and functional assays shed light on that *il11* is a potent regenerative factor for heart repair. During normal heart regeneration, *il11a* is produced from non-CMs at the early phase to trigger regenerative programs by initiating cell cycle reentry for CMs, activating epicardium for generating EPCs and stimulating vascularization. Its expression is decommissioned at the late regenerative phase to block aberrant outcomes, such as preventing fibrosis resolution. We demonstrate that *il11a* induction in the uninjured heart can stimulate regeneration programs, indicating that *il11* is a potent regenerative factor to enhance heart repair. Notably, we showed that *il11a* induction can enlarge hearts via CM hyperplasia rather than hypertrophy. However, prolonged epicardium activation by *il11a* leads to the persistent presence of EPCs, resulting in the constant production of cardiac fibroblasts and subsequent fibrosis. Additionally, our analysis of scRNA-seq profiles obtained from an extensive number of cardiac cells[44] revealed widespread expression of *il11ra*, a zebrafish homolog of *il11* receptor gene, across multiple cardiac cell types, including CMs and epicardium, supporting the pivotal function of IL11 signaling in CMs and epicardium (Supplementary Fig. 11c). Overall, our study demonstrates that *il11* signaling should be tightly regulated for effective and successful heart regeneration.

Regenerative paracrine factors are promising therapeutic candidates, due to their potential efficacy as drugs and simple delivery methods. Cytosolic and nuclear proteins require specific methodologies to be delivered inside the cells, presenting an additional challenge[72–74]. However, our studies reveal that application of paracrine factors for heart regeneration should be carefully considered, especially in the aspect of cardiac fibrosis. One essential yet paradoxical regenerative event is to accumulate ECM proteins, such as collagen, by cardiac fibroblasts at the injury site. Although this fibrotic response is thought to be detrimental due to its non-contractile feature potentially causing heart failure[75], it is indispensable and beneficial for heart regeneration at the early phase[13,76,77]. As demonstrated in our study, a single paracrine factor and signaling can target multiple cardiac cells, including CMs, endocardium/endothelium, and epicardium. Notably, the activated epicardial cells undergo reprogramming to generate progenitor cells, which later differentiate into the cardiac fibroblasts contributing to fibrosis. Consequently, a paracrine factor can serve as a double-edged sword to induce beneficial regenerative effects as well as detrimental cardiac fibrosis. In nature, these two effects are precisely balanced through tight spatiotemporal control during heart regeneration. However, current methodologies to enhance heart regeneration often face limitations, such as targeting multiple cell types by viral vectors or delivery vehicles[78], and obscure duration for functional recovery[79]. Here, we prove that delivering *il11a* paracrine factor, along with inhibiting cardiac fibrosis via ERK inhibitor treatment, in injured hearts can maintain CM proliferation without exacerbating fibrosis. Therefore, paracrine factor treatment combined with attenuating cardiac fibrosis can represent a more effective regenerative medicine strategy.

Previous works on *il11* across animal species and tissues have revealed seemingly conflicting results, as *il11* is known to have both regenerative and fibrotic properties[18–23,30,80–85]. Regarding the hearts, cardioprotective and fibrotic roles of *il11* have been reported[21–23,30,84,85]. Both *il11ra* mutant (loss-of-function for IL11 signaling)[21] and *il11*OE (gain-of-function in this work and ref. 85)[85] models exhibited cardiac fibrosis phenotypes. Loss-of-function study demonstrated that *il11ra* mutants exhibit an elevated number of myofibroblasts derived from endocardium and epicardium, leading to defective resolution of cardiac fibrosis. Tissue-specific rescue experiment of *il11ra* mitigates endocardium-derived myofibroblast differentiation, but not epicardium, suggesting the key roles of *il11* signaling in the endocardium for endothelial-to-mesenchymal transition (EndoMT). Our gain-of-function study revealed that *il11a*-mediated fibrotic effect is indirect through prolonged activation of epicardium. *il11*-induced epicardial activation is crucial to establish a regenerative niche at the initial phase of regeneration. However, persistent epicardium activation by *il11* results in continuous generation of cardiac fibroblasts, ultimately leading to cardiac fibrosis. These findings can explain controversies of *il11* signaling by highlighting pleiotropic function of *il11* depending on the location and duration of its expression.

Our study offers important new insights into IL11 biology for cardiac regeneration and fibrosis. A deeper understanding of IL11-associated cellular and molecular processes will be critical for developing a therapeutic strategy for precise regenerative effects with limited fibrosis.

## Methods

### Zebrafish

Wild-type or transgenic male and female zebrafish of the outbred Ekkwill (EK) strain ranging up to 18 months of age were used for all zebrafish experiments. The following transgenic lines were used in this study: *Tg(cmlc2:creER)^pd10*, *Tg(cmlc2:mCherry-N-2A-Fluc)^pd71*, *Tg(col12 a1b:EGFP)^wcm108*, *Tg(fli1:EGFP)^y1*, *Tg(−6.5kdrl:mCherry)^ci5*, *Tg(tcf21:m Cherry-NTR)^pd108*, *hapln1b*:EGFP[4,49,50,86–89]. To generate *actb2:loxP-BFP-loxP-il11a* (*uwk28*), the transgenic construct was generated by modifying *actb2:loxP-BFP-loxP-nrg1*[90]. The *nrg1* sequence were replaced with *il11a* cDNA amplified by PCR using wild-type zebrafish cDNA libraries. Primers are listed in Supplementary Table 1. The entire cassette was flanked with I-*Sce*I sites for meganuclease-enhanced transgenesis. The identified founder line was crossed with *cmlc2:creER* to induce *il11a* induction upon tamoxifen treatment. Water temperature was maintained at 26 °C for animals unless otherwise indicated. Work with zebrafish was performed in accordance with University of Wisconsin–Madison guidelines. Partial ventricular resention surgery was performed as described previously[1], in which ~20% of the cardiac ventricle was removed at the apex. Ventricular cryoinjuries were performed as described[91]. Briefly, the ventricular wall was applied by a cryoprobe precooled in liquid nitrogen.

### Generation of *il11a*^EGFP

To generate *il11a* knock-in reporter (*uwk25*), we generated a donor construct having 5′ and 3′ homology arms (HAs) and *EGFP-SV40pA* cassette. 5′ and 3′ HAs were amplified using genomic DNA extracted from fish that were used for injection. These HA fragments were subcloned into the pCS2-EGFP-pA-IsceI vector. For linearization, sgRNA located in 3′ HA ("GG ATC AAGTG TTACT CGCTC AGG") was used. sgRNAs synthesis and microinjection were described in ref. 92. Briefly, sgRNA were synthesized by a cloning-free method using the MEGAshortscript T7 Transcription Kit (Invitrogen, AM1354) and purified by the RNA purification Kit (Zymogen, R1016) according to the manufacturer's instructions. A sgRNA (25-30 ng/μl) and a donor plasmid (20−25 ng/μl) were mixed and co-injected with Cas9 protein (0.5 μg/μl; PNABio, CP01) into the one-cell stage embryos. After injection, none of larvae exhibit EGFP expression without injury. To sort F0 animals carrying *EGFP-pA* at the *il11a* locus, fin folds were amputated at 3 or 4 dpf, and embryos displaying EGFP fluorescence near the injury site at 1 dpa

were selected. EGFP-positive larvae were raised to adulthood, and founders were screened with $F_1$ progenies driving fin fold injury-inducible EGFP expression. Integration at the *il11a* locus was identified by PCR with primers described in Supplementary Table 1, followed by Sanger sequencing. For expression pattern to determine spatio-temporal expression of *il11a*[EGFP], at least three biological replicates, unless indicated, were examined per each time point and per observation to allow for potential variability within a single experiment.

## Immunostaining, histology and imaging

Hearts were fixed with 4% paraformaldehyde (PFA) for 1 h at room temperature. Cryosectioning and immunohistochemistry were performed as described previously with modifications[25]. Hearts were cryo-sectioned at 10 μm. Heart sections were equally distributed onto four to six serial slides so that each slide contained sections representing all area of the ventricle. 5 or 10% goat serum, 1% BSA, 1% DMSO, and 0.2% Triton X-100 solution was used for blocking and antibody staining. The primary and secondary antibodies used in this study were: anti-myosin heavy chain (mouse, F59; Developmental Studies Hybridoma Bank; 1:50 or mouse, MF20; Developmental Studies Hybridoma Bank; 1:50), anti-EGFP (rabbit, A11122; Life Technologies; 1:200), anti-EGFP (chicken, GFP-1020; Aves Labs; 1:2000), anti-Ds-Red (rabbit, 632496; Clontech; 1:500), anti-mCherry (chicken, MC87977980; Aves Labs; 1:500), anti-Raldh2 (rabbit, GTX124302; Genetex; 1:200), anti-PCNA (mouse, P8825; Sigma; 1:200), anti-α-Actinin (mouse, A7811; Sigma, 1:200), anti-ACTA2 (rabbit, GTX124505; GeneTex; 1:200), anti-pERK (rabbit, 9101; Cell Signaling Technology; 1:250). Anti-Mef2 (rabbit; 1:200)[49], Anti-phospho-Histone H3 (rabbit, 9701; Cell Signaling Technology, 1:100), Anti-phospho STAT3 (rabbit, 9131S; Cell Signaling Technology, 1:100), anti-Periostin (rabbit, 19899-1-AP, Proteintech, 1:200), Alexa Fluor 488 (mouse, rabbit, and chicken; A11029, A11034, and A11039; Life Technologies; 1:500), Alexa Fluor 594 (mouse and rabbit; A11032 and A11037; Life Technologies; 1:500). TUNEL staining was performed using in situ cell death detection kit (11684795910, Roche) in immunostaining. ISH on cryosections of 4% paraformaldehyde-fixed hearts was performed as previously described[25]. To generate digoxigenin-labeled probes, we used cDNA for 3′ part of each gene. Primer sequences are listed in Supplementary Table 1. Images of cardiac tissue sections were acquired using an Eclipse Ti-U inverted compound microscope (Nikon) or BZ-X810 fluorescence microscope (Keyence). Images were processed using either NIS-Elements (Nikon), ZEN (Zeiss), BZ-X800 analyzer (Keyence) or FIJI/ImageJ software.

## AFOG staining and quantification of collagen deposition

AFOG staining was performed on 10 μm sections as described[1]. Samples were first incubated at 60 °C for 2 h with Bouin preheated at 60 °C for 30 min; slides were then rinsed in running water for 30 min, and incubated with phosphomolybdic acid solution, followed by the sample incubation in AFGO staining solution for 10 min. For quantification of collagen deposition, collagen⁺ area was manually measured using ImageJ and normalized by the ventricular size and normalized by the entire ventricle size. Two to four largest sections were analyzed for each heart.

## Drug treatment

For tamoxifen treatments, 6-month to 1-year-old adult zebrafish were bathed in 5 μM tamoxifen (Sigma-Aldrich, T5648) for 24 h. zebrafish were rinsed with fresh aquarium water and returned to the recirculating water system for feeding. For 4-hydroxytamoxifen (4-HT) treatments, 6-month to 1-year-old adult zebrafish were intraperitoneally injected with 10 μl of 3 mM 4-HT (Sigma-Aldrich, H7904) for 4 consecutive days and returned to the water system as described previously[93]. For EdU treatments, 10 mM EdU was intraperitoneally injected once every 24 h for three consecutive days 24 h prior to heart collection. For U0126 treatments, Fish were intraperitoneally injected

with 10 μl of the 10 μM U0126 or 0.1 % DMSO in phosphate-buffered saline (PBS) using a 10 μl Hamilton syringe (1701SN) once daily for 30 days. CM ablation and imaging were performed as previously described in ref. 94. Briefly, 5 dpf *cmlc2:mCherry-NTR* zebrafish were incubated in 10 mM Metronidazole for 18 h. The Mtz-containing medium was replaced with fresh embryo medium several times and larvae were returned to 28 °C.

## Whole mount imaging

Whole mount imaging was performed as described in ref. 95. Hearts were isolated from tricaine-anesthetized adult animals. Freshly isolated hearts were rinsed with 1X PBS and immediately mounted in 1% low-melting agarose/PBS (0.01% tricaine) in a glass bottom dish. The images were acquired using an Eclipse Ti-U inverted compound microscope (Nikon). Confocal sections covering the entire heart were imaged from ventral sides and processed to obtain maximal intensity projections. *Fli1a:EGFP*⁺ area within a ventricle was assessed using ImageJ/Fiji image to measure vascular density.

## Quantification of ventricular size

Hearts were collected from tricaine-anesthetized adult animals and fixed with 4% PFA for 1 h at room temperature. Hearts were rinsed with 1X PBS with 0.1% Tween-20 and aligned on specimen stage. The images were acquired using a Zeiss Axio Zoom.V16 fluorescence stereo microscope. For quantification of ventricular size, the size was measured as described in ref. 32. the ventricular outline was manually traced, and the area encompassed by the outline was quantified using ZEN microscopy software (Blue edition).

## Quantification of cardiomyocyte size

The size of cardiomyocytes was quantified as previously described in ref. 32. Zebrafish ventricles were fixed in 3% PFA for 5 min and incubated in PBS supplemented with 1 mg/mL collagenase type 4 (Worthington Biochemical) at 4 °C overnight. Dissociated cardiac muscle cells are deposited into a slide using a cytocentrifuge (Thermo Scientific, Cytospin 4), followed by immunofluorescence using an anti-α-Actinin antibody. Images of α-Actinin⁺ cells were acquired by BZ-X810 fluorescence microscope (Keyence) and used to measure the size of individual cardiomyocytes using FIJI/ImageJ software.

## RNA isolation and bioinformatics

RNA was isolated from control and *il11a*OE hearts using Tri-Reagent (ThermoFisher). Complementary DNA (cDNA) was synthesized from 300 ng to 1 μg of total RNA using a NEB ProtoScript II first strand cDNA synthesis kit (NEB, E6560). For RNA-sequencing, total RNA was prepared from three biological replicate pools of 7 dpt or 7 mpt uninjured heart samples of control and *il11a*OE transgenic fish. Generation of mRNA libraries and sequencing were performed at the Biotechnology center at UW-Madison using an Illumina NovaSeq with 150 bp paired-end runs. Sequences were aligned to the zebrafish genome (GRCz11) using HISAT2[96]. Differentially regulated transcripts were identified using Featurecount[97] and Deseq2[98]. GO-term and GSEA analyses were done by the enrichGo and gseGO functions of clusterProfiler[99]. Accession numbers for transcriptome data sets are GSE233833.

scRNA-seq analysis of uninjured and regenerating hearts was done as described in ref. 100. Briefly, we obtained count files from GSE159032 and GSE158919[44] and reanalyzed with Seurat package[101]. Low quality cells (nFeature_RNA ≥ 4100, mitoRatio ≥ 0.25) were filtered out. The 50 principal components (PCs) of the PCA with resolution 4 were used for clustering. Marker genes to identify cell-types are listed in ref. 100.

## Quantification of Mef2+ cells

Quantification of cardiomyocytes number in the ventricle was performed using the FIJI/ImageJ software. The entire ventricle region was

determined and selected as a ROI with guidance provided by the green channel (α-Actinin) images. To count the cardiomyocytes, red channel (Mef2) was converted to 8 bits in selected ROIs, followed by automatic count of the red circular objects with the Analyze Particles tool. The total numbers of cardiomyocytes in 5 serial cardiac sections were calculated.

## Quantification

Quantification of EGFP$^+$ area was performed using the FIJI/ImageJ software. 2–5 sections were used to determine the values in one cardiac sample. ROIs were determined by manual selection of heart outlines, with guidance provided by the red channel (MHC) images. ROIs were selected by 400 μm ($H$) × 400 μm ($W$) from apex of a ventricle in the uninjured hearts or from amputation plane in the injured hearts. To quantify EGFP$^+$, selected ROIs were converted to 8 bit, adjusted with threshold, which is determined by autofluorescence level of EGFP$^+$ areas in ROIs were evaluated. For CM proliferation index, PCNA$^+$ Mef2$^+$ CMs were quantified from Mef2$^+$ CMs as described previously[11]. Images of the injury area were taken by 400 μm ($W$) × 200 μm ($H$) from the apical edge of the ventricular myocardium. The number of Mef2$^+$ PCNA$^+$ cells were manually counted using FIJI/ImageJ software. Three or four sections were analyzed for each heart. Quantification in uninjured hearts was conducted similarly, except that images of the mid-ventricular myocardium were taken for quantification. For quantification of sarcomere disorganization, 3 α-Actinin$^+$ areas with 200 μm ($W$) × 200 μm ($H$) were randomly taken from one section in the ventricular myocardium. The disorganized sarcomere area was manually measured by drawing a ROI without sarcomeric striation pattern. The sum of the ROIs was normalized by the entire α-Actinin$^+$ area using FIJI/ImageJ software. For quantification of cardiomyocyte density, at least three area (200 μm ($W$) × 200 μm ($H$)) were randomly taken from the ventricular myocardium. The number of Mef2$^+$ cells were manually counted using FIJI/ImageJ software. For quantification of pERK$^+$ or Postn$^+$ area in *tcf21*:mCherry$^+$ area, ROIs were determined by red channel (mCherry) images and the green$^+$ (pERK or Postn$^+$) area in ROIs were evaluated. Statistical values and *p*-values are described in each figure legend.

## Reporting summary

Further information on research design is available in the Nature Portfolio Reporting Summary linked to this article.

## Data availability

Data associated with this study are presented in the paper or in the Supplementary Materials. RNA-sequencing data are deposited in GEO under the accession code GSE233833. Source data are provided in the Source Data file. Source data are provided with this paper.

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

## Acknowledgements

We thank UW-Madison School of Medicine and Public Health (SMPH) BRMS (Biomedical Research Models Services) staffs for zebrafish care; the University of Wisconsin Biotechnology Center DNA Sequencing Facility (Research Resource Identifier – RRID:SCR_017759) for providing RNA-seq services; Seungwon Ryu for assistance with sample preparation; and Jeffery F. Dilworth for comments on the manuscript. Fig. 4j was created in BioRender.com. This work was supported by National Institutes of Health (R35GM137878, R01HL151522, and P30CA014520 to J.K.; R01HL155607 and R01HL166518 to J.C.; and R01HL142762 to J.W.), University of Wisconsin Institute for Clinical and Translational Research (UW ICTR) pilot grant to J.K., Stem Cell and Regenerative Medicine Center Research Training Award to K.S., and Hilldale undergraduate fellowship to I.M.S.

## Author contributions

Biological experiments: K.S., A.R., C.K., Y.X., J.S.; Computational analysis and data curation: K.S., C.D., I.M.S., J.K.; Conceptualization: K.S., J.K.; Writing original draft: K.S., J.K.; Reviewing, Editing: K.S., J.K.; Supervision: S.K., J.C., J.K.; Funding: J.C., J.K., J.W.

## Competing interests

The authors declare no competing interests.
