## [Transparent Peer Review file · Nature Communications]

Harnessing the regenerative potential of interleukin11 to enhance heart repair

Corresponding Author: Dr Junsu Kang

A version of this paper was originally rejected for publication by Nature Communications, however that decision was reconsidered after appeal by the authors.

Version 0:

Reviewer comments:

Reviewer #1

(Remarks to the Author)

In this manuscript, Shin et al. investigate the role of the cytokine Interleukin-11 in heart regeneration and fibrosis using a gain of function approach in zebrafish. They find that ectopic overexpression of one of two zebrafish orthologs of IL-11, il11a, in cardiomyocytes activates the expression of regeneration related processes. These include cardiomyocyte proliferation and the expansion of the coronary vasculature, even in the absence of experimental cardiac injury. However, prolonged ectopic expression of il11a in uninjured hearts lead to the emergence of collagen depositions, which the authors interpret as fibrosis. They show that while il11-dependent ERK activation can be observed in epicardial cells, it is absent from cardiomyocytes, suggesting that these two effects can be uncoupled. To test this, the authors devise a combinatorial treatment that involves il11a overexpression paired with pharmacological ERK inhibition. This approach preserved the enhanced CM proliferation but ameliorated collagen deposition, possibly as Erk activation occurs exclusively in epicardial cells after il11a overexpression while it could not be detected in cardiomyocytes.

The data is of high quality and the topic is certainly interesting, as there is some unresolved controversy in the addressed field. Il-11 has recently been shown to promote fibrosis in mammals through non-canonical ERK/mTor activation in several organs. These findings have sparked the development of promising Il11 inactivating drugs for therapeutic use possibly soon going into clinical trials. However, Il11 has also been identified as pro regenerative in mammals and essential for regeneration in regenerative species like tadpoles and zebrafish. Here, interleukin-11 function was shown to activate a regenerative gene program and to prevent fibrotic remodelling.

While the novelty of the results is somewhat limited by the fact that both, fibrotic effects through erk and regenerative effects through stat3 have previously been linked to Il11 signalling in loss of function studies this is the first study to provide results from a gain of function approach in a regenerative model hence allowing important comparisons to similar experiments carried out in mammals.

Nevertheless, the manuscript has some problems with consistencies and premature conclusions that need to be addressed thoroughly.

Concerns:

1. Exclusion of tissue damage by il11a overexpression

The authors claim that overexpression of il11a in cardiomyocytes using CRE activation alone results in the transcriptional activation of a regenerative gene program at early stages and fibrosis at late stages. However, they do not provide evidence that IL-11 overexpression does not lead to cardiac tissue damage, e.g. cell death, which could trigger these responses indirectly. The authors need to rule out that overexpression induces cell death at early and late stages (7 days, 3 months) triggering il11a independent events. IL-11 induced cell death has been reported in the liver for example (Anissa A. Widjaja et al., 2021)

2. Experimental design not adequately explained.

The authors use tamoxifen to ectopically express il11a in cardiomyocytes after recombination. However, the authors do not adequately explain the experimental design as they do not give the precise stage of the animal at which recombination was induced. This is critical because overexpression during early developmental stages (e.g. conversion in larvae) might cause developmental heart defects resulting in more severe il11a independent (indirect) downstream effects. For any experiment the precise time of conversion should be given.

3. Regeneration vs fibrosis

The authors state that fibrosis (a pathology term) is a component of the regenerative response which is confusing (e.g. page 3 or page 8)

*page3 – ‘The regenerative events in CMs are orchestrated by molecular signals provided by activated non-muscular cells, including epicardium, endocardium, and immune cells, which create a regenerative niche. Furthermore, similar to injured mammalian hearts, zebrafish exhibit an acute response to cardiac injury by forming fibrotic tissue at the injury site.’

It is unclear how cardiac fibroblasts can generate a regenerative niche and fibrosis at the same time.

As a consequence, it remains unclear whether the authors refer to ‘transient fibrosis’ or ‘pathological fibrosis’ several times throughout the manuscript. E.g. When col12 expression in pro regenerative EPCs is part of the regenerative program as indicated on (page 7; Fig 3 E), why is collagen deposition at the same location considered fibrotic (Fig.4 A). The authors should more precisely discriminate between these two conditions in the text. Furthermore, the authors need to provide more experimental evidence for pathological fibrosis when referring to the same. This includes experimental evidence for pathological fibrosis marker gene expression (e.g. COL1, SOX9, TGFB, POSTN) in il11OE animals in addition to the indiscriminate collagen staining in figure 4A. ( also see 4. regarding SMA expression). These experiments can be carried out by rt-PCRs and/or IHC.

4. Myofibroblast differentiation as a hallmark of fibrosis

Figure 4C is a pivotal figure as it resembles the only data provided that aims to assess ACTA2/SMA+ myofibroblast abundance and hence pathological fibrosis directly. However, the provided image does not show overlapping expression of SMA and tcf21 as expected for epicardial derived myofibroblasts (see Figure 3E or 4E for overlapping expression), suggesting the depicted SMA+ cells are not myofibroblasts. SMA is also expressed in perivascular smooth muscle cells which could be more abundant in the il11aOE hearts (hypervascularization phenotype). Importantly, SMA is also known to be upregulated in dedifferentiating cardiomyocytes or in cardiomyocytes of failing hearts (reviewed by Yike Zhu and Roger Foo, 2021). In addition, il11a was previously described to limit myofibroblast differentiation in zebrafish (Allanki et al, 2021). Therefore, the authors need to include a counterstain for endothelial cells and particularly cardiomyocytes to confirm that the cells depicted are not CMs. Alternatively, providing direct evidence through lineage tracing of epicardial cells under il11aOE conditions would effectively address this concern.

5. Statistical analysis

The authors have not always included numbers on the frequency of the observed phenotypes. This limits the reader's ability to assess the experiment's replication rate and how representative the images are. The authors should add the according numbers wherever no statistical analysis is provided e.g. Fig 3E.

6. Activation of STAT signalling in cardiomyocytes

IL-11 is an activator of the JAK STAT, ERK and the AKT pathway. The authors state on page 5 and in Fig S2 that ‘il11a induction also highly upregulated key components of janus kinase/signal transducers and activators of transcription (JAK/STAT) signaling pathway, including jak1, stat3, and socs3b, which constitute a canonical downstream pathway of IL-11’. The authors conclude that ‘JAK/STAT pathway is a major downstream effectors of il11a in CMs’. However, all three pathways work independent of a transcriptional regulation of their components but by phosphorylation cascades leading to active p-STAT and p-ERK. The authors should test whether STAT3 gets activated by IL-11 overexpression by assessing pSTAT levels instead of RNA levels of kinases which could very well be inactive. This would be particularly interesting as published data suggests that IL-11 exclusively activates ERK in mammals (Schafer S. et al, 2017).

Additional Questions:

- Do the authors consider the observed cardiomyocytes proliferation in Il11OE animals direct or indirect by secreted factors in the epicardial cells?
- Is the Interleukin-11 receptor gene expressed by cardiomyocytes?
- Are proliferating cardiomyocytes p-STAT3 positive? (Fig. S6)
- What do the authors mean with ‘injury-responsible transcription factor complex AP-1 components’ (page 5)

Reviewer #2

(Remarks to the Author)

The authors present an impressive body of work demonstrating that overexpression of IL11a is sufficient to induce cardiomyocyte hyperplasia in uninjured zebrafish hearts, and can enhance CM proliferation during heart regeneration. However, it also induces fibrosis; the authors manage to uncouple these processes via inhibition of ERK signaling. The data are generally of very high quality (with a few exceptions pointed out below). Identification of factors that are sufficient to induce CM proliferation *in vivo* is rather rare, thus the paper is certainly interesting and noteworthy. However, considering that the necessity for IL11 for heart regeneration and CM proliferation has been shown previously, we think that in order to be suitable for Nat Comm, the manuscript would have to be expanded to answer one or several of the following questions:

1) Endogenous IL11a appears to represent one of the signals that mediates the effect of non-CMs on CM proliferation. The authors here somewhat surprisingly find that CM-specific overexpression conversely has profound effects on the epicardium and endothelium. Have the authors also attempted epicardium- and/or endothelium-specific overexpression?

2) Sven Reischauer's lab has shown that il11ra mutants have reduced CM proliferation and increased fibrosis. How do the authors reconcile this with their finding that il11a overexpression likewise induces fibrosis?

3) An important novelty of the manuscript is that the authors argue that they found a regime that enhances CM proliferation without inducing fibrosis using the combined overexpression of il11a and inhibition of ERK signaling. Yet, by conventional readouts for the overall "success of regeneration" used in the fish field, the combined il11a OE + ERK inhibition would still count as failed (compared to controls), since the fish present scars. Other works have also indicated that CM proliferation and scarring can be uncoupled (eg. Bertozzi et al., Dev Biol. 2021 Mar;471:106-118.). Can the authors find other readouts of regeneration that show that il11a OE + ERK inhibition does at least in some aspects better than unperturbed regeneration? E.g. functional assays, number of cardiomyocytes etc.).

In addition to expansion towards deeper mechanistic insight along these or other lines, the authors should also address the following issues:

Major:

1. Fig. 1B: it remains unclear in which region the GFP fluorescence was measured. y-axis title states "apex of ventricle", figure legend "near the wound area", and the methods don't contain information on which region was selected and how this was done in an unbiased manner.
2. Fig. 2: the zebrafish heart regeneration field is guilty of using the term "cardiomyocyte proliferation" in an oversimplified manner. In mammals, it is very obvious that markers of cell cycling, like PCNA, or S-phase, like nucleotide incorporation, cannot be used to imply "proliferation" in cardiomyocytes. Even mitosis markers do not necessarily indicate successful cytokinesis. Affairs are easier in zebrafish, since generally there is a consensus that most if not all cycling CMs will also undergo mitosis. Yet, authors must not describe their results on increased PCNA staining after il11a overexpression as evidence for "induction of CM proliferation" or that it would be proof of "the mitogenic roles of il11a". While they also go on to show that cardiomyocyte density increases, authors should nevertheless show that il11a overexpression induces phospho-H3 positive CMs, both in noninjured and injured conditions. Y-axis labels for the PCNA data in Fig. 2D and Fig. 5B should be changed from "CM proliferation index" to e.g. "PCNA+ CMs (%)". Descriptions of PCNA data in the text should not refer to "proliferation" but rather to "cycling" e.g. in Fig. S6 legend.
3. Fig 3: is a bit bold to claim that il11a can stimulate the epicardium to generate EPCs which subsequently differentiate into EPDCs impacting coronary vascularization. This is only based on tcf21 and col12a1b staining. But, several populations of epicardial progenitors have been identified for example the hapln1a+ epicardial cells which help with regeneration. More evidence could be given, with in situ/HCR to maybe address in which transient population of epicardial cells IL11a has the effect.
4. Fig. 4B: it is unclear what was measured here. The figure heading reads "thickness of cortical layer", but the figure legend "quantification of collagen+ layer thickness". Neither can't be correct. While the cortical muscle layer is clearly identifiable in the controls, it appears incorrect to assume that the collagen+ area in the IL11a OE represents cortical muscle. Conversely, since no collagen is visible in the controls, how can the thickness of the collagen layer be measured there and give a non-zero results? Please clarify.
5. page 9: "The phosphorylated ERK (pERK) signal, representing active ERK, was minimal in the tcf21+ epicardium of the control hearts, but remarkably increased upon il11a induction". This conclusion makes it sound as if the level of pERK within the tcf21+ cells would increase, yet the pERK positive area appears to be increased because there are more tcf21+ cells; the authors don't show that pERK levels increase per cell. Please rephrase.
6. Fig. 5: collagen quantification: methods state that the measurements of the collagen+ area was done on "two to four largest sections". It appears that this could result in bias, depending on where within the ventricle the apical resection happened, which might not always be centered around the true apex. Thus, such quantifications should always be performed on an unbiased subset of sections of the entire heart, eg. every xth section.

Minor:

1. S1A scheme: the scheme is a bit unclear, since the 5'UTR must be part of an Exon, an the arrow which is usually is used to indicate the transcriptional start site here seem to indicate the start of the coding sequence. Please adapt to more clearly indicate where the exons are and within those the 5'UTR and coding sequence.
2. S3 and S4 should be reversed, since S4 is mentioned earlier in the text.

Reviewer #3

(Remarks to the Author)

In this study, Shin et al. investigate the role of interleukin-11a (IL11a) in heart regeneration in zebrafish. Using an EGFP knock-in (KI) reporter line, the authors characterized the dynamic expression of IL11a in epicardial and endocardial cells

during regeneration. Conditional over-expression (OE) of the IL11a gene in the uninjured heart increased sarcomere disassembly and cell cycle re-entry in cardiomyocytes (CMs), and the heart enlarged over time. IL11a-OE also increased coronary vessels in the cortical layer of the ventricle with the emergence of the epicardial-derived col12a1b+ progenitor cells. Consistent with the previous reports indicating the role in fibrosis, IL11a-OE resulted in the collagenous tissue formation in both homeostasis and regeneration contexts, and the authors proposed to enhance heart repair by inducing IL11a expression with inhibiting its fibrotic function through ERK inhibition.

The manuscript was written well, and the figures were presented clearly with excellent images. I particularly liked that the authors generated genetic tools to investigate the expression and function of IL11a. Overall, the study was nicely done, but my primary concern about this study is the lack of novelty.

My specific comments are as follows:

- 1) The analysis of IL11a gene expression with the new KI reporter line (Fig. 1), although beautifully shown, does not seem to add new information to the published result (Fang, Y. et al., Proc Natl Acad Sci USA 110:13416-2, 2013).
- 2) The authors concluded that IL11a-OE induced cardiomegaly via CM hyperplasia, not hypertrophy (p. 6, last sentence), but the evidence supporting this conclusion seems weak. In the IL11a-OE heart, the CM number was slightly increased (Fig. 2I), but the width of CMs also showed an increasing trend. The authors' conclusion may be too simplified, and the data may suggest that IL11a-OE could induce CM hypertrophy and hyperplasia both at a low level.
- 3) It is unclear how the authors could quantify disassembled sarcomeres in tissue sections, as shown in Fig. 2E.
- 4) The authors attributed the increased vascularization observed within the lateral wall of the IL11a-OE ventricle to the epicardial progenitor cells (EPCs) marked by col12a1b-EGFP, which expanded in the same ventricle area (Fig. 3E). The conclusion would be far more convincing if the authors could provide evidence to support the vasculogenic function of the col12a1b+ EPCs.
- 5) Similarly, the authors would need evidence to support the pro-fibrosis function of ACTA2+ cardiac fibroblasts (Fig. 4C).
- 6) ACTA2+ cells and col12a1b-EGFP+ cells are both derived from the epicardium. I'm wondering whether these cell populations are distinct and whether the authors examined the overlaps of these markers.
- 7) A recent paper reported the result of myocardial over-expression of IL11 in mice (Sweeney, M. et al., Int J Mol Sci 24:12989, 2023). This paper should be cited and discussed in the manuscript.

Version 1:

Reviewer comments:

Reviewer #1

(Remarks to the Author)

1. The Shin et al. manuscript comes in a somewhat improved version after some of the concerns were addressed, however, some of my major concerns have unfortunately not been addressed with the necessary rigor.

In mammals, it was shown that IL11 promotes fibrosis (e.g. Schafer et al, 2017). In zebrafish, it was shown that IL11 is essential for tissue regeneration, limiting fibrosis (e.g. Tsujioka et al., 2017, Allanki et al., 2021). While there are several possibilities why that could be the case, including postulated differences resulting from evolution (Cook JofCTR 2023), the specific strength of the work from Shin et al. lies in its gain of function approach in zebrafish potentially being able to contribute to help clarifying this controversy.

The central assertion of the manuscript is that IL11 overexpression in zebrafish leads to fibrosis, akin to observations in mammalian models. However, this claim remains to lack sufficient empirical support. While the authors have refined their language to some extent to clarify that regeneration involves transient extracellular matrix (ECM) deposition which is obviously different from pathological fibrosis, they assert that cardiac IL11 overexpression results in fibrosis without adequately demonstrating the upregulation of fibrosis markers widely recognized in mammalian studies and presented in the zebrafish knockout study (e.g. Allanki et al. Sci Adv 2021).

In the referenced study, a loss of IL11 signalling was associated with a broad activation of a fibrotic gene program, including essential regulators of fibrosis such as TGF1/2, SOX9, EGR1, and ECM cross-linking enzymes like LOX, which distinguish fibrotic from regenerative ECM deposition. In addition it was shown that loss of IL11 signalling leads to a differentiation of ACTA2+ myofibroblasts from both the epicardial and endocardial layers. Notably, myofibroblast differentiation was shown to be mitigated by reactivating IL11 signalling in a cell type-specific manner, affecting only endocardial but not epicardial cells. Hence in my previous review, I recommended directly assessing the fibrosis-specific markers to clearly differentiate between regenerative ECM remodelling and fibrosis. However, the authors have decided to limit their analysis and to only present a limited range of ECM genes, including various collagens (and FN1) that are also upregulated during regeneration, and thus are not unique to pathological types of fibrosis.

Furthermore, the origin of the observed ACTA2+ cells in the IL11 overexpression animals at late time points remains unclear. The authors have not been able to provide evidence that these cells are derived from lineages representing the

canonical progenitors of cardiac fibrosis seen in mammals. The authors performed additional immunostaining to observe that these ACTA2+ cells were not cardiomyocytes, not endothelial cells, and not epicardial derived cells. Hence, they could as well be other ACTA2+ cell types including SMC, or, based on their anatomical location (in between cortical and trabecular cardiomyocyte layers) they could be primordial layer cardiomyocytes, which were previously shown to be negative for sarcomeric structures (Tadesse-Tsedeke et al, Dev Biol 2021). This ambiguity could be resolved by co-staining with embCMHC or MHC (MF20) or HABP. If indeed these ACTA2+ cells are primordial layer CMs, it would mean that IL11 overexpression induces CM dedifferentiation, which is a regenerative response – quite the opposite of fibrosis. In line with this, the authors have already shown in Fig S2 that IL11 OE hearts indeed display increased CM dedifferentiation. Hence, I strongly recommend the authors check their bulk RNA-seq both at 7 dpt and 7 mpt for the fibrotic markers mentioned above, and CM dediff markers, e.g. those published in Tadesse-Tsedeke et al 2021. In addition, in Fig S9e, the authors show increased ACTA2+ cells in IL11 OE hearts. They look more like vessel-associated SMCs. A co-staining with endothelial markers at this time point would be helpful.

From my perspective further characterizing these ACTA2+ cells and the fibrotic gene program appears paramount in calling it a fibrotic response.

In summary, I believe that the manuscript's central claim is not sufficiently substantiated by the necessary data. Substantial further investigation into the specific markers of fibrosis appears crucial to robustly support the conclusions drawn. Alternatively, the authors should substantially moderate their claim.

In addition, all related to the above.

2. In line 206 and 207 the authors write: “ACTA2 is used as a marker for vascular smooth muscle cells (VSMCs) and dedifferentiating CMs in addition to cardiac fibroblasts” – That is not correct. ACTA2 is not a marker for cardiac fibroblasts but for myofibroblasts. Quiescent cardiac fibroblasts do not express SMA/ACTA2 under physiological conditions. Furthermore, SMA/ACTA2 is not exclusive to fibroblast derived myofibroblasts but is also activated in other lineages (eg. Endothelial derived myofibroblasts). To discriminate myofibroblasts from SMCs, additional coexpression analysis for MYLK or SOX9 might help.

3. Related to the above, the authors claim in Line 251-252 of the revised manuscript that “Our data indicate that, while *il11a* induction gradually accumulates collagen in uninjured hearts, **its fibrotic function is accelerated in the presence of injury.**” – This claim ignores that genetic LOF data has been established in zebrafish providing evidence that *il11a* is essential for regeneration and limits fibrosis after injury. In addition, the authors show that COL12 reporter activation (Fig S9) and CM dedifferentiation (Fig S2) – both established regenerative responses – are exacerbated in the IL11 OE hearts. And the authors showed an increasing vascularization (also a regenerative response) of the IL11 OE ventricles without injury (Fig 3a-d). In line with this, the increased ACTA2+ cells in Fig S9e could be SMCs, which would imply an increased revascularization after injury as well. All these data (COL12 reporter, CM dediff, and revasc) suggests that these IL11 OE hearts are stuck in a continuous regenerative program possibly inducing progressive heart failure. In its current form, the manuscript does not contain sufficient data allowing to discriminate between these conditions.

4. 322-326 “Both *il11ra* mutant (loss-of-function for IL11 signaling) and *il11OE* (gain-of-function in this work and Ref. 75) models exhibited cardiac fibrosis phenotypes. **But their underlying mechanisms of fibrosis differ fundamentally. IL11ra mutant hearts lost the regenerative ability, leading to impaired function. This can cause cardiac dysfunction, thereby inducing pathological fibrosis as a compensatory mechanism.**”

- According to the reference, IL11 mutants show a robust activation of a broad fibrotic gene program, as early as 96 hours post cardiac injury. Hence, fibrosis in this case is very unlikely to be attributed to heart failure resulting from impaired regeneration. Furthermore, no impaired heart function or even heart failure was reported in the cited work. Notably, the same argument falls back on this manuscript as the ECM deposition described here as “fibrosis after IL11 overexpression” could be a consequence from the globally and continuously dedifferentiating cardiomyocytes “**leading to impaired heart function. This can cause cardiac dysfunction, thereby inducing pathological fibrosis as a compensatory mechanism.**”

– Given the extended period of time it takes IL11OE in zebrafish to induce significant ECM deposition compared to *il11ra* LOF induced fibrosis, the authors should possibly more deeply think about this argument in the light of their own conclusions.

Reviewer #2

(Remarks to the Author)

The authors have addressed all the specific issues that we have raised in a satisfactory manner. The overall quality of the data is high and the conclusions drawn are sufficiently backed by data.

However, in our review we pointed out that we thought that additional mechanistic insights might be needed to increase the overall novelty of the findings. Unfortunately, the authors were not able to add much along those lines. We appreciate that the authors made efforts; unfortunately, the first suggestion we made (overexpression in epicardium or endocardium) could not be done using the existing tools, since the Cre responder is not expressed in these cells. The second point was only met with a rather general discussion. For the third, the authors do provide additional evidence that the IL OE plus ERK inhibition does slightly increase CM numbers over unperturbed (vehicle treated wild-type) regeneration. Yet, as a potentially “therapeutic” approach this is not very impressive.

Thus, whether the paper presents enough novelty for Nat Comm is for the editors to decide. The existing data are

certainly solid and the conclusions well supported.

Reviewer #3

(Remarks to the Author)

I appreciate the authors' effort to address my comments. I am still awkward to agree with the conclusion in the revised manuscript: "Collectively, these results demonstrate that il11a activation can bypass the injury signal to induce majorly myocardial hyperplasia, highlighting the mitogenic effect of il11a in zebrafish hearts." (p. 7, lines 139-141.) To me, the data shown in Figure 2 suggests a very moderate proliferative effect of IL11 overexpression, and "majorly" doesn't seem to fit with the data. Because of this moderate phenotype, I also feel that the authors should consider reducing the tone about the mitogenic role of IL11 in other part of the manuscript.

Version 2:

Reviewer comments:

Reviewer #1

(Remarks to the Author)

In this revised version, Shin et al. provide additional data intending to add experimental support to previous claims and to correct incorrect statements in the previous version.

In short:

Background: Shin et al. use an overexpression line to dissect possible roles for Interleukin-11 (IL-11), an interleukin-6 family cytokine previously linked to regeneration and fibrosis in various experimental systems, most importantly in mouse, rat, and zebrafish hearts. While genetic loss- and gain-of-function data in mice revealed a profibrotic function for IL-11 in mice (by promoting ACTA2+ myofibroblast differentiation), genetic data in zebrafish revealed a crucial function for IL-11 to promote regeneration (limiting myofibroblast differentiation). In rats, systemic IL-11 administration showed cardioprotective effects, and rhIL-11 is an FDA-approved drug.

Study: The authors use a gain-of-function approach by conditionally overexpressing IL-11 in cardiomyocytes of zebrafish to shed light on the question of whether excessive IL-11 promotes regeneration or fibrosis. Previously, the authors showed that aspects of a regenerative response, including hypervascularization and increased cardiomyocyte proliferation, are induced by IL-11 overexpression at early stages close to the activation of the IL-11 overexpression. These findings were in line with previous reports that IL-11 acts as a pro-regenerative cytokine in zebrafish and other models. The authors further claimed that prolonged expression results in fibrosis based on the identification of ACTA2+ cells, which were interpreted as myofibroblasts (a hallmark of fibrosis), and ECM deposition. However, the origin and nature of the ACTA2+ cells remained unclear, and the detected ECM deposition cannot be interpreted as maladaptive remodeling without further evidence, as ECM deposition is also a hallmark of regeneration.

In this revised version of the manuscript, the authors now partially address previous concerns and substantiate some of the claims.

New results:

The authors now add data showing that IL-11 overexpression results in delayed Periostin expression in tcf21 positive cells, a marker for activated cells in the context of regeneration and fibrosis in zebrafish (Figure 4 e, d, f, g). As with the previous analysis of other markers, Periostin is not a fibrosis-exclusive marker but is also expressed during regeneration, leaving this analysis on its own insufficient to draw strong conclusions. This problem becomes bigger as the canonical hallmark of cardiac fibrosis, the differentiation of ACTA2+ myofibroblasts from cardiac stromal cells, could not be validated during revisions but turned out to be dedifferentiating cardiomyocytes.

However, the extended period of time of 30 days after activation of the overexpression suggests that Periostin expression in this case (30+ days) is not a direct consequence of IL-11 expression but rather of secondary effects, supporting the notion that Periostin expression can be regarded as a consequence of maladaptive cardiac remodeling.

Hence, the authors now use RNAseq data to investigate the dynamics of regeneration and fibrosis in their system using a fibrotic and regenerative marker gene set at different stages after IL-11 activation. This experiment now shows that even seven days after transgene activation, none of the fibrotic markers is activated through IL-11 overexpression, but rather the regeneration-related gene set shows activation. Only at seven months post-transgene activation (7 mpt), enhanced gene expression of fibrotic genes was detected, including periostin expression.

Taken together, while previously published results for early and possibly direct activation of regenerative genes could be established, the prolonged time to induce fibrotic gene set expression rather speaks for an indirect effect

of IL-11, possibly caused by organ failure. The causes could result from dedifferentiation and excessive growth of the myocardial layer, but the analysis is speculative at this point.

I am still hesitant to attribute fibrosis directly to IL-11 overexpression in this model exclusively based on the activation of a fibrotic gene set and Periostin expression only after 30 days to three months, with no detectable emergence of myofibroblasts and only after clear dedifferentiation of cardiomyocytes occurred. In my view, the authors should clearly discuss the possibility that this is a secondary, indirect effect, as now also included in the text section related to point 4.

The authors now clearly state that “Our gain-of-function study revealed that il11a-mediated fibrotic effect is indirect through prolonged activation of epicardium” in the revised text. I believe this is a fair interpretation of the results and also compatible with the observed gene expression and Periostin activation, and should be the general interpretation of the work rather than a direct consequence. However, this remains inconsistent in the current version of the manuscript, e.g., Line 215 (“Given that the fibrosis driven by il11a...”) should be “Given that the fibrosis observed in il11a overexpressing hearts” or Line 225: “il11a-induced fibrosis” should be “fibrosis resulting from ectopic il11a expression” and more.

Of note: The authors write “Tissue-specific rescue experiment of il11ra mitigates endocardium-derived myofibroblast differentiation, but not epicardium, suggesting the key roles of il11 signaling in the endocardium for endothelial-to-mesenchymal transition (EndoMT).” To my understanding, in the cited experiment, il11ra was exclusively rescued in the endocardium, which sufficiently explains why myofibroblast differentiation was only affected in these cells but not in the epicardium. The fact that a global loss of il11ra leads to an increase in myofibroblast differentiation in both the endocardium and epicardium strongly suggests key roles in both cell types.

Reviewer #2

(Remarks to the Author)

In my response to the revisions I had pointed out that I had found it somewhat disappointing that the authors had not been able to add more data to the manuscript that would more clearly illustrate its advance over existing literature. With this second revision they have now expanded on the issue of how their finding that il11a overexpression induces fibrosis fits with the published results that il11ra mutants likewise present with fibrosis. I also appreciate the more detailed discussion and requests for additional data that reviewer 1 has had with the authors along those lines.

Overall, I find that the paper is further strengthened. The main message, which in my view is that the complicated pro- and anti-regenerative roles of il11 are not only seen in the poorly regenerative mammalian hearts, but also in highly regenerative zebrafish, is worth reporting in Nat Comms.

Reviewer #3

(Remarks to the Author)

I appreciate the revision, which seems more accurate in describing the results.

Reviewer #1 (Remarks to the Author):

In this manuscript, Shin et al. investigate the role of the cytokine Interleukin-11 in heart regeneration and fibrosis using a gain of function approach in zebrafish. They find that ectopic overexpression of one of two zebrafish orthologs of IL-11, *il11a*, in cardiomyocytes activates the expression of regeneration related processes. These include cardiomyocyte proliferation and the expansion of the coronary vasculature, even in the absence of experimental cardiac injury. However, prolonged ectopic expression of *il11a* in uninjured hearts lead to the emergence of collagen depositions, which the authors interpret as fibrosis. They show that while *il11*-dependent ERK activation can be observed in epicardial cells, it is absent from cardiomyocytes, suggesting that these two effects can be uncoupled. To test this, the authors devise a combinatorial treatment that involves *il11a* overexpression paired with pharmacological ERK inhibition. This approach preserved the enhanced CM proliferation but ameliorated collagen deposition, possibly as Erk activation occurs exclusively in epicardial cells after *il11a* overexpression while it could not be detected in cardiomyocytes.

The data is of high quality and the topic is certainly interesting, as there is some unresolved controversy in the addressed field. IL-11 has recently been shown to promote fibrosis in mammals through non-canonical ERK/mTor activation in several organs. These findings have sparked the development of promising IL11 inactivating drugs for therapeutic use possibly soon going into clinical trials. However, IL11 has also been identified as pro regenerative in mammals and essential for regeneration in regenerative species like tadpoles and zebrafish. Here, interleukin-11 function was shown to activate a regenerative gene program and to prevent fibrotic remodelling.

While the novelty of the results is somewhat limited by the fact that both, fibrotic effects through erk and regenerative effects through stat3 have previously been linked to IL11 signalling in loss of function studies this is the first study to provide results from a gain of function approach in a regenerative model hence allowing important comparisons to similar experiments carried out in mammals.

Nevertheless, the manuscript has some problems with consistencies and premature conclusions that need to be addressed thoroughly.

Reply: We thank the reviewer for the thoughtful review and very kind words as well as noticing specific issues we overlooked.

Concerns:

1. Exclusion of tissue damage by *il11a* overexpression

The authors claim that overexpression of *il11a* in cardiomyocytes using CRE activation alone results in the transcriptional activation of a regenerative gene program at early stages and fibrosis at late stages. However, they do not provide evidence that IL-11 overexpression does not lead to cardiac tissue damage, e.g. cell death, which could trigger these responses indirectly. The authors need to rule out that overexpression induces cell death at early and late stages (7 days, 3 months) triggering *il11a* independent events. IL-11 induced cell death has been reported in the liver for example (Anissa A. Widjaja et al., 2021)

Reply: We appreciate the reviewer for the important comment on the potential effect of *il11a* overexpression (OE). We agree that *il11a*OE may cause cardiac tissue damage or cell death,

potentially leading to regeneration response. To address it, we have performed Terminal deoxynucleotidyl transferase (TdT) dUTP Nick-End Labeling (TUNEL) assay with control and *il11a*OE hearts at 10 days post-treatment (dpt) and 3 months post-treatment (mpt) of 4-hydroxytamoxifen (4-HT). We demonstrated that TUNEL+ cells were rarely detected in both 10 dpt and 3 mpt *il11a*OE hearts while TUNEL+ cells were obviously detectable in positive control samples with cryo-injured hearts at 7 dpci (days post-cryoinjury).

These new results are included in Supplementary Fig. S3.

We have added new text on Page 5.

Page 5

il11a-triggered CM proliferation could be caused by cell death or damages. To test this possibility, we performed Terminal deoxynucleotidyl transferase (TdT) dUTP Nick-End Labeling (TUNEL) assay and found that TUNEL+ cells are undetectable in *il11a*OE hearts, while our control experiment with cryoinjured hearts displayed robust TUNEL+ cells in 7 dpci (days post-cryoinjury) hearts (**Supplementary Fig. 3**). Our results indicate that *il11a*OE unlikely induces apoptosis but directly triggers CM proliferation.

2. Experimental design not adequately explained.

The authors use tamoxifen to ectopically express *il11a* in cardiomyocytes after recombination. However, the authors do not adequately explain the experimental design as they do not give the precise stage of the animal at which recombination was induced. This is critical because overexpression during early developmental stages (e.g. conversion in larvae) might cause developmental heart defects resulting in more severe *il11a* independent (indirect) downstream effects. For any experiment the precise time of conversion should be given.

Reply: We apologize for the lack of explanation about our experiments. For *il11a* overexpression, we used 6-month to 1-year-old adult fish for our tamoxifen (Tam) and 4-hydroxytamoxifen (4-HT) experiments. As we induced the *il11a*OE in the adult fish but not larvae, we can rule out the influence of *il11a*OE on development. In the revised manuscript, we edited our method to clarify the precise stage of the fish.

Method, Page 17

Drug treatment

For tamoxifen treatments, 6-month to 1-year-old adult zebrafish were bathed in 5 μ M tamoxifen (Sigma-Aldrich, T5648) for 24 hr. zebrafish were rinsed with fresh aquarium water and returned to the recirculating water system for feeding. For 4-hydroxytamoxifen (4-HT) treatments, 6-month to 1-year-old zebrafish were intraperitoneally injected with 10 μ l of 3mM 4-HT (Sigma-Aldrich, H7904) for 4 consecutive days and returned to the water system as described previously

3. Regeneration vs fibrosis

The authors state that fibrosis (a pathology term) is a component of the regenerative response which is confusing (e.g. page 3 or page 8)

*page3 – ‘The regenerative events in CMs are orchestrated by molecular signals provided by

activated non-muscular cells, including epicardium, endocardium, and immune cells, which create a regenerative niche. Furthermore, similar to injured mammalian hearts, zebrafish exhibit an acute response to cardiac injury by forming fibrotic tissue at the injury site.'

It is unclear how cardiac fibroblasts can generate a regenerative niche and fibrosis at the same time.

As a consequence, it remains unclear whether the authors refer to 'transient fibrosis' or 'pathological fibrosis' several times throughout the manuscript. E.g. When *col12* expression in pro regenerative EPCs is part of the regenerative program as indicated on (page 7; Fig 3 E), why is collagen deposition at the same location considered fibrotic (Fig.4 A). The authors should more precisely discriminate between these two conditions in the text. Furthermore, the authors need to provide more experimental evidence for pathological fibrosis when referring to the same. This includes experimental evidence for pathological fibrosis marker gene expression (e.g. COL1, SOX9, TGFB, POSTN) in *il11aOE* animals in addition to the indiscriminate collagen staining in figure 4A. ( also see 4. regarding SMA expression). These experiments can be carried out by rt-PCRs and/or IHC.

Reply: We thank the reviewer for the thoughtful comment. We agree that transient fibrosis and pathological fibrosis should be discriminated more concisely in our text to avoid confusion. To address this comment, we have changed the term of regenerative fibrosis to "regenerative ECM" or "ECM remodeling". We have added "pathological" fibrosis for non-regenerative fibrosis in the text.

We have also clarified the occurrence of pathological fibrosis in *il11aOE* by performing additional experiments. To provide more comprehensive evidence, we have performed bulk RNA-sequencing analysis with 7 months post-Tam treatment (mpt) control and *il11aOE* heart. Then, we compared the results between 7 days post-treatment (dpt) RNA-seq and 7 mpt RNA seq. Our analysis revealed that pathological fibrosis marker genes, such as *postnb*, *col1a1a*, *col1a1b*, are significantly upregulated in 7 mpt *il11aOE* hearts, whereas those genes are not induced in 7 dpt *il11aOE* hearts, compared to control. Moreover, we performed *in situ* hybridization (ISH) to detect *postnb* transcripts in 3 mpt hearts. Our data show that *postnb* expression is robustly upregulated in the cortical layer of *il11aOE* hearts at 3 mpt, but not in control.

These new results are included in Supplementary Fig. 7 and Supplementary Data 2.

We have added new text on Pages 3 and 10.

Page 3

Furthermore, similar to injured mammalian hearts, zebrafish exhibit an acute response to cardiac injury by **depositing extracellular matrix (ECM) proteins** at the injury site.

Page 9

Given recent reports indicating ... **Pathological fibrosis is evident by expression of pathological fibrotic genes, including *col1a1a*, *col1a1b*, *col5a1*, *col6a3*, and *col14a1a*, primarily produced by cardiac fibroblasts^{51, 52, 53, 54}. To investigate whether these genes are induced by long-term *il11aOE*, we performed RNA-seq analysis with 7 mpt control and *il11aOE* hearts. Our transcriptome analysis revealed a significant upregulation of pathological fibrotic and cardiac fibroblast marker genes in 7 mpt, but not 7 dpt, *il11aOE* hearts (**Supplementary Fig. 7a and Supplementary Data 2**). We also observed strong induction of *postnb* in the epicardial and cortical layer of 3 mpt *il11aOE* hearts (**Supplementary Fig. 7b**), indicating the fibrotic effect of**

il11 in zebrafish.

4. Myofibroblast differentiation as a hallmark of fibrosis

Figure 4C is a pivotal figure as it resembles the only data provided that aims to assess ACTA2/SMA⁺ myofibroblast abundance and hence pathological fibrosis directly. However, the provided image does not show overlapping expression of SMA and *tcf21* as expected for epicardial derived myofibroblasts (see Figure 3E or 4E for overlapping expression), suggesting the depicted SMA⁺ cells are not myofibroblasts. SMA is also expressed in perivascular smooth muscle cells which could be more abundant in the *il11a*OE hearts (hypervascularization phenotype). Importantly, SMA is also known to be upregulated in dedifferentiating cardiomyocytes or in cardiomyocytes of failing hearts (reviewed by Yike Zhu and Roger Foo, 2021). In addition, *il11a* was previously described to limit myofibroblast differentiation in zebrafish (Allanki et al, 2021). Therefore, the authors need to include a counterstain for endothelial cells and particularly cardiomyocytes to confirm that the cells depicted are not CMs. Alternatively, providing direct evidence through lineage tracing of epicardial cells under *il11a*OE conditions would effectively address this concern.

Reply: We appreciate the reviewer for the thoughtful suggestion. To address this comment, we have performed immunostaining with cardiac sections of 1 mpt *il11a*OE hearts and compared the localization of ACTA2(SMA)⁺ cells with *fli1a*⁺ endothelial cells or alpha-actinin⁺ cardiomyocytes (CMs). Our data show that ACTA2⁺ cells are localized close to *fli1a*⁺ endothelium in control, indicating that ACTA2⁺ cells are perivascular smooth muscle cells in control hearts. However, we found that the majority of ACTA2⁺ cells are not associated with *fli1a*⁺ endothelial coronary vessels in *il11a*OE. We also examined whether ACTA2⁺ cells are co-labelled with CM marker in *il11a*OE hearts. However, we observed ACTA2⁺ CMs are highly limited and the majority of ACTA2⁺ cells appear not to be CMs.

The new results are included in Supplementary Fig. 7.

We have added the explanation of the experiment on Page 10.

Page 10

In regenerating zebrafish hearts ... ACTA2 is used as a marker for vascular smooth muscle cells (VSMCs) and dedifferentiating CMs in addition to cardiac fibroblasts^{57, 58}. To discriminate ACTA2-expressing cells, we examined their location and colocalization with blood vessels and CM markers. VSMCs are ACTA2⁺ cells surrounding *fli1*⁺ coronary vessels in the cortical layers. In control hearts, ACTA2⁺ cells were found aligning with *fli1a:EGFP*⁺ coronary vessels in the outer cortical layer of ventricles (**Supplementary Fig. 7c**). Although circular ACTA2⁺ VSMC layers were also evident in *il11a*OE hearts, we found that *il11a*OE hearts additionally display a thick and linear ACTA2⁺ layer in the deeper cortical layer (**Supplementary Fig. 7c**). Also, these ACTA2⁺ layers are not closely associated with *fli1a:EGFP*⁺ cells, and the majority of these ACTA2⁺ cells lacks a CM marker (**Supplementary Fig. 7c-e**). Thus, our results indicate that *il11a*-induced ACTA2⁺ layers in the deeper cortical layer are not VSMCs or dedifferentiating CMs, but cardiac fibroblasts, such as myofibroblasts.

5. Statistical analysis

The authors have not always included numbers on the frequency of the observed phenotypes. This limits the reader's ability to assess the experiment's replication rate and how representative

the images are. The authors should add the according numbers wherever no statistical analysis is provided e.g. Fig 3E.

Reply: To address the reviewer's concern, we have added how many replicates have been used for the individual experiment in each figure legend.

6. Activation of STAT signalling in cardiomyocytes

IL-11 is an activator of the JAK/STAT, ERK and the AKT pathway. The authors state on page 5 and in Fig S2 that 'il11a induction also highly upregulated key components of janus kinase/signal transducers and activators of transcription (JAK/STAT) signaling pathway, including jak1, stat3, and socs3b, which constitute a canonical downstream pathway of IL-11'. The authors conclude that 'JAK/STAT pathway is a major downstream effectors of il11a in CMs'.

However, all three pathways work independent of a transcriptional regulation of their components but by phosphorylation cascades leading to active p-STAT and p-ERK. The authors should test whether STAT3 gets activated by IL-11 overexpression by assessing pSTAT levels instead of RNA levels of kinases which could very well be inactive. This would be particularly interesting as published data suggests that IL-11 exclusively activates ERK in mammals (Schafer S. et al, 2017).

Reply: We thank the reviewer for the suggestion. We acknowledged that visualization of p-STAT level provides an alternative approach to assess the JAK/STAT activation by *il11a*OE. To address this comment, we have performed the immunostaining with Mef2 (CM nuclei marker) and pSTAT3 antibodies on cardiac sections of 10 dpt control and *il11a*OE hearts. While Mef2+ CMs are unlikely p-STAT3 positive in the control, ~40 % of CMs in the *il11a*OE hearts are p-STAT3 positive. Thus, our data suggest that *il11a*OE can activate the JAK/STAT pathway as a downstream effector in CM.

These new results are included in Supplementary Fig. 4.

We have added these results in the text on Page 6.

Page 6

... Moreover, *il11a*OE robustly increased phospho-STAT3 (p-STAT3) labelled CMs and 5.87% of pSTAT-3⁺ Mef2⁺ CMs are labelled by EdU, suggesting the active STAT3 pathway in CMs (Supplementary Fig. 4c, d).

Additional Questions:

-Do the authors consider the observed cardiomyocytes proliferation in *il11a*OE animals direct or indirect by secreted factors in the epicardial cells?

Reply: We consider that *il11a*OE-induced CM proliferation is mainly caused by the direct effect of *il11a* in CMs. Our immunostaining and RNA-seq analyses demonstrated that the CM proliferation effect is robust and prompt right after *il11a*OE, as shown with 7 dpt *il11a*OE hearts. This CM proliferation likely proceeds to *il11a*-mediated epicardial activation. As shown in Fig 3G, *il11a*-mediated epicardial activation appears to start by 10 dpt, when a minimum number of *col12a1b*⁺ epicardium emerges in the cardiac surface. Mesenchymal epicardial cells representing epicardial progenitor cells are strongly detectable in the cortical inner layer by 30

dpt. Although some CM proliferation can be achieved indirectly by secreted factors from the epicardium, we expect the *il11a* can induce CM proliferation directly via JAK/STAT pathway.

-Is the Interleukin-11 receptor gene expressed by cardiomyocytes?

Reply: We thank the reviewer for raising the question regarding the expression of the *il11a* receptor, *il11ra*. To investigate the *il11ra* expression in the zebrafish hearts, we reanalyzed the published single cell RNA-seq data. Our analysis identified that *il11ra*, a cognate *il11a* receptor, is expressed in diverse cell types, including CMs and epicardium, in the regenerating hearts, such as 3 and 7 dpa (days post-amputation). These results also support that *il11* signaling can activate the downstream signaling in the CMs.

We have included this data in Supplementary Fig. 10c.

We have added the text on Page 13

Page 13

.... Additionally, our analysis of scRNA-seq profiles obtained from an extensive number of cardiac cells⁴⁴ revealed widespread expression of *il11ra*, a zebrafish homolog of *il11* receptor gene, across multiple cardiac cell types, including CMs and epicardium, supporting the pivotal function of IL11 signaling in CMs and epicardium (**Supplementary Fig. 10c**). ...

-Are proliferating cardiomyocytes p-STAT3 positive? (Fig. S6)

Reply: Our new data with p-STAT3 and Mef2 staining approach identified that ~6% of p-STAT3+ Mef+ CMs are EdU+, suggesting noticeable portion of p-STAT3+ CMs are proliferative.

We have added these new results in Supplementary Fi. 4.

-What do the authors mean with 'injury-responsible transcription factor complex AP-1 components' (page 5)

Reply: We apologize for the confusion in the text. We attempted to use injury-induced AP-1 transcription factor as an example of regeneration-associated factor. AP-1 components are well-known injury-inducible or regeneration-dependent transcription factor. However, we noticed that injury-responsible can cause readers to misunderstand our explanation. To avoid this, we have deleted 'injury-responsible transcription factor complex' in the sentence.

Reviewer #2 (Remarks to the Author):

The authors present an impressive body of work demonstrating that overexpression of IL11a is sufficient to induce cardiomyocyte hyperplasia in uninjured zebrafish hearts, and can enhance CM proliferation during heart regeneration. However, it also induces fibrosis; the authors manage to uncouple these processes via inhibition of ERK signaling. The data are generally of very high quality (with a few exceptions pointed out below). Identification of factors that are sufficient to induce CM proliferation in vivo is rather rare, thus the paper is certainly interesting and noteworthy. However, considering that the necessity for IL11 for heart regeneration and CM

proliferation has been shown previously, we think that in order to be suitable for Nat Comm, the manuscript would have to be expanded to answer one or several of the following questions:

Reply: We are pleased to know that the reviewer considers our work as an interesting area. We also sincerely appreciate the reviewer for the thoughtful comments.

1) Endogenous IL11a appears to represent one of the signals that mediates the effect of non-CMs on CM proliferation. The authors here somewhat surprisingly find that CM-specific overexpression conversely has profound effects on the epicardium and endothelium. Have the authors also attempted epicardium- and/or endothelium-specific overexpression?

Reply: We appreciate the reviewer for making a thoughtful suggestion. To address the reviewer's comments, we crossed *actb2:loxp-BFP-loxp-il11a* with *tcf21:creER* or *kdrl:creER* for epicardium- or endocardium-specific *il11a* overexpression (*il11aOE*). We administered 4-HT with young adults and collected hearts at 30 days post tamoxifen-treatment (dpt) or 8 month post tamoxifen-treatment (mpt) to assess the fibrotic effect of *il11a*. Additionally, 10 dpt hearts were collected to examine pro-regenerative effect of epicardium- and endocardium-derived *il11a* induction. However, we did not observe enhanced CM proliferation and fibrosis in epicardium- and endocardium-derived *il11aOE* hearts, compared to control (Please see figure1).

actb2 was considered as a ubiquitously expressed gene as it is a housekeeping gene. Therefore, the *actb2* promoter fragment was considered as a ubiquitous promoter and used with a *loxp* construct. Previous studies utilizing the *actb2*-linked *loxp* alleles can successfully drive gene expression with several *CreER* line, such as *cmhc2:CreER*. However, the *actb2* promoter fragment used in the field and our work is characterized as not genuinely ubiquitous as several cell types cannot express a gene linked with the *actb2* promoter. Additionally, there could be a line variation presumably due to the chromosomal effect of the integration site. To investigate whether the *actb2* promoter sequence linked with the *loxp-BFP-loxp-il11a* cassette can direct epicardium and endocardium expression, we evaluated *acta2*-directed *BFP* expression with *tcf21:mCherry⁺*, an epicardium marker, and *fli1a:EGFP⁺*, endothelium marker. Our analysis demonstrated that *BFP*-expressing cells unlikely label both epicardium and endocardium (Please see figure2). These results indicate that the current *Tg(actb2:loxp-BFP-loxp-il11a)* allele would not enable to overexpress *il11a* in the epicardium and endocardium. Due to this technical

challenge, we could not address the reviewer's comment in the current manuscript, but we will address this interesting question in future following study by generating other gain-of-function allele.

2) Sven Reischauer's lab has shown that *il11ra* mutants have reduced CM proliferation and increased fibrosis. How do the authors reconcile this with their finding that *il11a* overexpression likewise induces fibrosis?

Reply: We appreciate the reviewer's insightful comment. Dr. Reischauer's lab showed that lack of *il11ra* reduced CM proliferation, similar to our hypothesis, but increased fibrosis in the regenerated hearts, which may be considered as opposite outcomes compared to ours. Although fibrotic outcomes are similar in our gain-of-function and their loss-of-function studies, the underlying molecular mechanisms would be fundamentally different. *il11a*-induced fibrosis is caused by constant activation of epicardial cells as it leads to constant production of cardiac fibrosis. In contrast, *il11ra* mutants cause fibrosis due to cardiac dysfunction by losing regenerative ability. A similar observation is found by the Hippo-Yap pathway as ectopic activation of Yap in fibroblasts caused fibrosis although *Yap* mutants resulted in fibrotic hearts.

We have added this explanation in Discussion on Page 15.

Page 15

Previous works on *il11* across animal species and tissues have revealed seemingly conflicting results, as *il11* is known to have both regenerative and fibrotic properties^{18, 19, 20, 21, 22, 23, 30, 70, 71, 72, 73, 74, 75}. Regarding the hearts, antifibrotic/cardioprotective and fibrotic roles of *il11* have been reported^{21, 22, 23, 30, 74, 75}. Both *il11ra* mutant (loss-of-function for IL11 signaling)²¹ and *il11OE* (gain-of-function in this work and Ref. 75)⁷⁵ models exhibited cardiac fibrosis phenotypes. But their underlying mechanisms of fibrosis differ fundamentally. *il11ra* mutant hearts lost the regenerative ability, leading to impaired function. This can cause cardiac dysfunction, thereby inducing pathological fibrosis as a compensatory mechanism. In contrast, *il11aOE* activates epicardial cells to induce ECM remodeling during the initial phase to establish a regenerative niche. However, prolonged activation of epicardium by *il11* results in constant generation of cardiac fibroblasts, ultimately leading to cardiac fibrosis. Similar outcomes regarding regenerative pathways are also

reported by the Hippo pathway. While *Yap1* conditional mutation in CMs of neonatal mice exhibits fibrotic hearts⁷⁶, ectopic activation of *Yap1* in cardiac fibroblasts also induces fibrosis^{77, 78, 79, 80}. Therefore, duration of regenerative factor treatment and their target cells should be considered to achieve successful heart repair without adverse outcomes.

3) An important novelty of the manuscript is that the authors argue that they found a regime that enhances CM proliferation without inducing fibrosis using the combined overexpression of *il11a* and inhibition of ERK signaling. Yet, by conventional readouts for the overall “success of regeneration” used in the fish field, the combined *il11a*OE + ERK inhibition would still count as failed (compared to controls), since the fish present scars. Other works have also indicated that CM proliferation and scarring can be uncoupled (eg. Bertozzi et al., Dev Biol. 2021 Mar;471:106-118.). Can the authors find other readouts of regeneration that show that *Il11a* OE + ERK inhibition does at least in some aspects better than unperturbed regeneration? E.g. functional assays, number of cardiomyocytes etc.).

Reply: We thank the reviewer for pointing out the area where we can make improvements. To address this comment, we evaluated the positive effect of *il11a* on heart regeneration by counting the number of CMs in the sequential 5 sections for each condition (control+DMSO, control+ERK inhibitor, *il11a*OE+DMSO, and *il11a*OE+ERK inhibitor). Our data show that *il11a*OE hearts have significantly more CMs in both DMSO- and ERK inhibitor-treated condition than control DMSO-treated hearts, indicating that combinatorial treatment can enhance cardiac regeneration.

The new results are included in the Supplementary Fig. 10.

We added the new text on Page 12.

Page 12

... As another readout of regeneration, we quantified the total number of *Mef2*⁺ CMs throughout a ventricle in 45 dpi hearts and found that U0126-treated *il11a*OE hearts have significantly more *Mef2*⁺ CMs than control hearts (**Supplementary Fig. 10a, b**). These results highlight the specific effect of ERK/MAPK inhibition on cardiac fibrosis with limited influence on regeneration.

In addition to expansion towards deeper mechanistic insight along these or other lines, the authors should also address the following issues:

Major:

1. Fig.1B: it remains unclear in which region the GFP fluorescence was measured. y-axis title states “apex of ventricle”, figure legend “near the wound area”, and the methods don’t contain information on which region was selected and how this was done in an unbiased manner.

Reply: We apologize for the lack of explanation about our experiments. To clarify the quantification method in Fig. 1b, we have revised the description in the method section.

Page 20

Quantification of EGFP⁺ area was performed using the FIJI/ImageJ software. 2-5 sections were used to determine the values in one cardiac sample. ROIs were determined by manual selection of heart outlines, with guidance provided by the red channel (MHC) images. ROIs were selected

by 400 μm (H) x 400 μm (W) from apex of a ventricle in the uninjured hearts or from amputation plane in the injured hearts.

2. Fig. 2: the zebrafish heart regeneration field is guilty of using the term “cardiomyocyte proliferation” in an oversimplified manner. In mammals, it is very obvious that markers of cell cycling, like PCNA, or S-phase, like nucleotide incorporation, cannot be used to imply “proliferation” in cardiomyocytes. Even mitosis markers do not necessarily indicate successful cytokinesis. Affairs are easier in zebrafish, since generally there is a consensus that most if not all cycling CMs will also undergo mitosis. Yet, authors must not describe their results on increased PCNA staining after *il11a* overexpression as evidence for “induction of CM proliferation” or that it would be proof of “the mitogenic roles of *il11a*”. While they also go on to show that cardiomyocyte density increases, authors should nevertheless show that *il11a* overexpression induces phospho-H3 positive CMs, both in noninjured and injured conditions. Y-axis labels for the PCNA data in Fig. 2D and Fig. 5B should be changed from “CM proliferation index” to e.g. “PCNA+ CMs (%)”. Descriptions of PCNA data in the text should not refer to “proliferation” but rather to “cycling” e.g. in Fig. S6 legend.

Reply: We appreciate the reviewer for the thoughtful comment. To address the reviewer’s comment, we investigated whether *il11a*OE can induce phospho-H3 CMs in the uninjured hearts by immunostaining. We noticed a significant increase in the level of pH3 in CMs in the uninjured *il11a*OE hearts, but control hearts exhibited extremely low level of pH3 in CMs.

We also examined pH3 CMs in injured hearts, but there is no significant difference between control and *il11a*OE hearts. As injured zebrafish hearts are proliferative due to the presence of *il11a* induction and percentage of pH3 positive cells are highly limited, compared to PCNA, pH3 difference would not be high enough to discriminate. However, our data clearly demonstrated that *il11a*OE can increase pH3 positive cells in uninjured hearts and more CM cells in 1-month post-treated hearts, suggesting mitogenic roles of *il11a* on CMs.

The new results are included in Supplementary Fig. 2.

We have added new text on Page 5.

Additionally, as the reviewer pointed out, we have changed not only Y-axis labels from “CM proliferation index” to “PCNA+ CM (%)”, but also the expression in the text from “cell proliferation” to “cell cycling”.

Page 5

... Strikingly, *il11a*OE led to the marked induction of cell cycle reentry in CM of the uninjured hearts, whereas control hearts showed little to no cell cycle activity in CMs (**Fig. 2b, d**). Moreover, we noted a significant induction of phospho-H3 positive CMs in the uninjured *il11a*OE hearts, compared to control (**Supplementary Fig. 2a, b**), providing evidence for the mitogenic roles of *il11a* on CMs. ...

Page 11

We next explored whether this dual functionality of *il11a* is also operational in injured hearts. At 14 days post-injured hearts, ectopic *il11a* activation significantly increased **CM cycling** ($8.61 \pm 1.26\%$ and $5.20 \pm 0.80\%$ in OE and control, respectively) (**Fig. 5a, b and Supplementary Fig. 8a**), confirming its mitogenic effect on injured hearts.

Page 12

Moreover, we observed that CMs in *il11a*OE hearts treated with U0126 and vehicle exhibited a similar **cell cycling** rate (**Fig. 5i, j**).

3. Fig 3: is a bit bold to claim that *il11a* can stimulate the epicardium to generate EPCs which subsequently differentiate into EPDCs impacting coronary vascularization. This is only based on *tcf21* and *col12a1b* staining. But, several populations of epicardial progenitors have been identified for example the *hapln1a*⁺ epicardial cells which help with regeneration. More evidence could be given, with in situs/HCR to maybe address in which transient population of epicardial cells *IL11a* has the effect.

Reply: We appreciate the reviewer for the comment. To address this comment, we investigated whether *hapln1b* cells labeling regenerative EPDCs emerge by *il11a*OE in the uninjured hearts. We observed the expansion of *hapln1b*⁺ cell populations in the cortical layer of *il11a*OE at 30 dpt. These data indicate that *il11a* induction can give rise to multiple EPDC sub-populations.

The new result was included in Fig. 3.

We added the new text on Pages 9.

Page 9

In contrast, *col12a1b*⁺ EPCs were conspicuously absent in control hearts at 10 and 30 dpt. **We also tested whether *il11a*OE can stimulate the activation of *hapln1b*⁺ EPDCs as *hapln1*⁺ EPDCs are known to impact coronary growth⁸. Our data revealed that *il11a*OE hearts display the expansion of *hapln1b*⁺ cell populations in the mesenchymal layer at 30 dpt (**Fig. 3h, i**).** Therefore, our results indicate that *il11a* can stimulate the epicardium to generate EPCs which subsequently differentiate into EPDCs impacting coronary vascularization.

4. Fig. 4B: it is unclear what was measured here. The figure heading reads “thickness of cortical layer”, but the figure legend “quantification of collagen+ layer thickness”. Neither can’t be correct. While the cortical muscle layer is clearly identifiable in the controls, it appears incorrect to assume that the collagen+ area in the *IL11a* OE represents cortical muscle. Conversely, since no collagen is visible in the controls, how can the thickness of the collagen layer be measured there and give a non-zero results? Please clarify.

Reply: We appreciate the reviewer’s comment. Collagen+ area does not accurately represent the cortical layer in *il11a*OE, potentially making readers confused. We re-quantified the thickness of collagen⁺ layer in both control and *il11a*OE and revised the figure annotation and the figure legend.

5. page 9: “The phosphorylated ERK (pERK) signal, representing active ERK, was minimal in the *tcf21*⁺ epicardium of the control hearts, but remarkably increased upon *il11a* induction”. This conclusion makes it sound as if the level of pERK within the *tcf21*⁺ cells would increase, yet the pERK positive area appears to be increased because there are more *tcf21*⁺ cells; the authors don’t show that pERK levels increase per cell. Please rephrase.

Reply: We apologize for overstating our findings. We rephrase our sentences.

... While control hearts contain a minimal number of phospho-ERK (pERK)⁺ *tcf21*⁺ epicardial cells, their numbers are significantly increased upon *il11a* induction (**Fig. 4e, f and Supplementary Fig. 7f, g**). ...

6. Fig. 5: collagen quantification: methods state that the measurements of the collagen⁺ area was done on “two to four largest sections”. It appears that this could result in bias, depending on where within the ventricle the apical resection happened, which might not always be centered around the true apex. Thus, such quantifications should always be performed on an unbiased subset of sections of the entire heart, eg. every xth section

Reply: To address this comment, we quantified collagen deposition from entire ventricular sections of control and *il11a*OE hearts at 30 days post-amputation (dpa). These new data are added in the Supplementary Figure 8b and c.

Minor:

1. S1A scheme: the scheme is a bit unclear, since the 5'UTR must be part of an Exon, and the arrow which is usually used to indicate the transcriptional start site here seem to indicate the start of the coding sequence. Please adapt to more clearly indicate where the exons are and within those the 5'UTR and coding sequence.

Reply: We apologize for the ambiguity of our scheme in the figure. To clarify the reviewer's point, we have deleted the arrow and indicated where the start codon is in the scheme.

2. S3 and S4 should be reversed, since S4 is mentioned earlier in the text.

Reply: We have corrected it.

Reviewer #3 (Remarks to the Author):

In this study, Shin et al. investigate the role of interleukin-11a (IL11a) in heart regeneration in zebrafish. Using an EGFP knock-in (KI) reporter line, the authors characterized the dynamic expression of IL11a in epicardial and endocardial cells during regeneration. Conditional over-expression (OE) of the IL11a gene in the uninjured heart increased sarcomere disassembly and cell cycle re-entry in cardiomyocytes (CMs), and the heart enlarged over time. IL11a-OE also increased coronary vessels in the cortical layer of the ventricle with the emergence of the epicardial-derived col12a1b⁺ progenitor cells. Consistent with the previous reports indicating the role in fibrosis, IL11a-OE resulted in the collagenous tissue formation in both homeostasis and regeneration contexts, and the authors proposed to enhance heart repair by inducing IL11a expression with inhibiting its fibrotic function through ERK inhibition.

The manuscript was written well, and the figures were presented clearly with excellent images. I particularly liked that the authors generated genetic tools to investigate the expression and function of IL11a. Overall, the study was nicely done, but my primary concern about this study is

the lack of novelty.

Reply: We sincerely appreciate the reviewer's effort and comments on our manuscript.

My specific comments are as follows:

1) The analysis of IL11a gene expression with the new KI reporter line (Fig. 1), although beautifully shown, does not seem to add new information to the published result (Fang, Y. et al., Proc Natl Acad Sci USA 110:13416-2, 2013).

Reply: We appreciate the reviewer's comments. Although previous work conducted by Fang et al., showed *il11a* expression upon cardiac injury, their *in situ* hybridization (ISH) approach unlikely provides a high resolution map of spatiotemporal *il11a* expression during heart regeneration. Based on reviewer's comment, we took advantage of knock-in reporter line and dissected *il11a* expressing cells during heart regeneration. We identified dynamic *il11a* expressing cell types at 3 and 7 dpa as epicardial *il11a* expression distant to the wound area was decreased over time while endocardial expression near the wound area remained strong by 7 dpa.

The new results are included in Supplementary Fig. S1d, e.

The revised texts are added on Page 4.

Page 4

... Additionally, our further analysis to dissect *il11a* expressing cells revealed that epicardial *il11a*^{EGFP} expression decreased, whereas endocardial expression at the wound area remained constant from 3 to 7 dpa. (**Supplementary Fig. 1d, e**). ...

2) The authors concluded that IL11a-OE induced cardiomegaly via CM hyperplasia, not hypertrophy (p. 6, last sentence), but the evidence supporting this conclusion seems weak. In the IL11a-OE heart, the CM number was slightly increased (Fig. 2I), but the width of CMs also showed an increasing trend. The authors' conclusion may be too simplified, and the data may suggest that IL11a-OE could induce CM hypertrophy and hyperplasia both at a low level.

Reply: We appreciate the reviewer for the comment. Our CM size measurement demonstrated that there is no significant difference of width between control and *il11a*OE hearts. Indeed, the length of CMs isolated from *il11a*OE is significantly shorter than that of control. As the reviewer indicates, *il11a*OE hearts contain some large sized CMs, a possible sign of hypertrophic effect. Although our data, such as proliferative CM assays and RNA-seq analysis, largely support hyperplasia effects of *il11*, we acknowledge the potential hypertrophic effect of *il11*. To address this concern, we have revised our text.

Page 7

... The number of Mef2⁺ nuclei in the ventricle significantly increased in *il11a*OE, compared to control (123.45 ± 8.71 and 107.45 ± 12.23 CMs / 4 × 10⁴ μm² for *il11a*OE and control, respectively) (**Fig. 2h, i**). Additionally, the individual CMs isolated from *il11a*OE hearts exhibited smaller length (83.65 ± 23.5 μm) than control (101.2 ± 23.2 μm) (**Fig. 2j, l**). While there is no notable difference in width between control and *il11a*OE CMs, *il11a*OE hearts likely contain some enlarged CMs, indicating a possible low level of hypertrophic effect (**Fig. 2j, k**). Collectively, these results

demonstrate that *il11a* activation can bypass the injury signal to induce **majorly** myocardial hyperplasia, highlighting the mitogenic effect of *il11a* in zebrafish hearts.

3) It is unclear how the authors could quantify disassembled sarcomeres in tissue sections, as shown in Fig. 2E.

Reply: We apologize for the lack of a description for our experiment. To clarify the quantification method, we added it and revised the text by explaining how we quantified the disassembled sarcomeres on Page 23.

Page 23

For quantification of sarcomere disorganization, 3 α -actinin⁺ areas with 200 μ m (W) x 200 μ m (H) were randomly taken from one section in the ventricular myocardium. The disorganized sarcomere area without sarcomeric striation pattern was manually measured by drawing a ROI. The sum of the ROIs was normalized by the entire α -actinin⁺ area using FIJI/ImageJ software.

4) The authors attributed the increased vascularization observed within the lateral wall of the *il11a*-OE ventricle to the epicardial progenitor cells (EPCs) marked by *col12a1b*-EGFP, which expanded in the same ventricle area (Fig. 3E). The conclusion would be far more convincing if the authors could provide evidence to support the vasculogenic function of the *col12a1b*⁺ EPCs.

Reply: We appreciate the reviewer's comment to improve our manuscript. To address this comment, we examined whether *il11a*OE can induce the expansion of *hapln1*⁺ EPCs as the pro-vasculogenic role of *hapln1*⁺ epicardial cells has been reported (Sun *et al.*, 2023, *Nature communications*). We demonstrated that *hapln1b*⁺ cells expand in the cortical layer of *il11a*OE hearts at 30 dpt, compared to control. Additionally, we performed ISH analysis to define expressing cell types of angiogenic factors in response to *il11a*OE. Our ISH data demonstrated that multiple angiogenic factors are highly upregulated in epicardial and cortical layers of *il11a*OE hearts, compared to control. These results provide evidence of vasculogenic function of *il11a*-induced EPCs.

The new results are included in Fig. 3.

The revised texts are added on Pages 8 and 9.

Page 8

... Our ISH analysis also validated that the angiogenic factors are highly upregulated in the epicardial and cortical layer in *il11a*OE, but not in control (**Fig. 3f**).. ...

Page 9

... We also tested whether *il11a*OE can stimulate the activation of *hapln1b*⁺ EPDCs as *hapln1*⁺ EPDCs are known to impact coronary growth⁸. Our data revealed that *il11a*OE hearts display the expansion of *hapln1b*⁺ cell populations in the mesenchymal layer at 30 dpt (**Fig. 3h, i**).. ...

5) Similarly, the authors would need evidence to support the pro-fibrosis function of ACTA2+ cardiac fibroblasts (Fig. 4C).

Reply: We appreciate the reviewer for this suggestion. It is challenging to investigate fibrotic function of *acta2* in zebrafish due to the technical issue. Creating and examining an *acta2* mutant line would require over a year. This process includes generating heterozygote mutants, propagating them, and obtaining homozygotes from heterozygote mating, with each generation requiring three months in zebrafish. Given this time frame, this is not feasible for revision. Moreover, recent mouse work revealed that *Acta2* mutation unlikely influences myofibroblast differentiation and cardiac function after cardiac injury (Li et al., JMCC, 2022) due to compensatory mechanisms to transcribe the non-*Acta2* actin isoform. This result suggests that *acta2* mutant fish might maintain the capability to generate myofibroblasts and consequently induce fibrosis.

The major cell-type contributing cardiac fibrosis in injured hearts is myofibroblasts differentiated from cardiac fibroblasts, with ACTA2 being a commonly used marker for myofibroblasts. Myofibroblasts in zebrafish are also reported in several papers (Koth et al., Development, 2020; Yu et al, Sci. Rep., 2018; González-Rosa et al., Dev. Biol., 2012). However, we are aware that ACTA2 can label three different cell types, including vascular smooth muscle cells (SMCs), dedifferentiated CMs, and myofibroblasts. To further discern emerging ACTA2⁺ cell-types in *il11a*OE hearts, we have performed immunostaining with cardiac sections of 1 mpt *il11a*OE hearts and compared the localization of ACTA2⁺ cells with *fli1a*⁺ endothelial cells or alpha-actinin⁺ CMs. Our data demonstrate that ACTA2⁺ cells are localized close to *fli1a*⁺ endothelium in control, indicating that ACTA2⁺ cells are perivascular SMCs in control hearts. However, we found that the majority of ACTA2⁺ cells are not associated with *fli1a*⁺ endothelial coronary vessels in *il11a*OE. We also examined whether ACTA2⁺ cells are co-labelled with a CM marker in *il11a*OE hearts. We observed ACTA2⁺ CMs are highly limited and the majority of ACTA2⁺ cells appear not to be CMs. This analysis provides additional evidence supporting that the ACTA2⁺ layer in *il11a*OE heart represents myofibroblasts that contribute cardiac fibroblasts.

The new results are included in Supplementary Fig. 7.

We have added new results on Page 10.

Page 10

In regenerating zebrafish hearts ... ACTA2 is used as a marker for vascular smooth muscle cells (VSMCs) and dedifferentiating CMs in addition to cardiac fibroblasts^{57, 58}. To discriminate ACTA2-expressing cells, we examined their location and colocalization with blood vessels and CM markers. VSMCs are ACTA2⁺ cells surrounding *fli1*⁺ coronary vessels in the cortical layers. In control hearts, ACTA2⁺ cells were found aligning with *fli1a:EGFP*⁺ coronary vessels in the outer cortical layer of ventricles (**Supplementary Fig. 7c**). Although circular ACTA2⁺ VSMC layers were also evident in *il11a*OE hearts, we found that *il11a*OE hearts additionally display a thick and linear ACTA2⁺ layer in the deeper cortical layer (**Supplementary Fig. 7c**). Also, these ACTA2⁺ layers are not closely associated with *fli1a:EGFP*⁺ cells, and the majority of these ACTA2⁺ cells lacks a CM marker (**Supplementary Fig. 7c-e**). Thus, our results indicate that *il11a*-induced ACTA2⁺ layers in the deeper cortical layer are not VSMCs or dedifferentiating CMs, but cardiac fibroblasts, such as myofibroblasts.

6) ACTA2⁺ cells and col12a1b-EGFP⁺ cells are both derived from the epicardium. I'm wondering whether these cell populations are distinct and whether the authors examined the overlaps of these markers.

Reply: We thank the reviewer for the interesting question. To address this question, we visualized ACTA2⁺ and col12a1b^{EGFP}⁺ cells in the injured hearts at 45 dpi. Although the majority of ACTA2⁺ and col12a1b^{EGFP}⁺ are distinct, they are localized closely to each other and some cells are co-labelled by ACTA2 and col12a1b^{EGFP} at the injured site, indicating that ACTA2⁺ cells may be originated from col12a1b^{EGFP}⁺ cells.

The new results are included in Supplementary Fig. 9.

We have revised the text on Page 12.

Page 12

... Furthermore, ectopic *il11a* induction led to the persistence of col12a1b⁺ EPCs and fibrotic tissues at the injury site in cryoinjured hearts (**Fig. S9A-D**). Similar to *il11aOE* uninjured hearts, we observed ACTA2⁺ layers at the wound area, suggesting the existence of cardiac fibroblasts induced by *il11aOE*. The co-expression of col12a1b^{EGFP} and ACTA2 in some cells at the wound site suggests their origin from the activated EPCs (**Supplementary Fig. 9e**). Our data indicate that, while *il11a* induction gradually accumulates collagen in uninjured hearts, its fibrotic function is accelerated in the presence of injury.

7) A recent paper reported the result of myocardial over-expression of IL11 in mice (Sweeney, M. et al., Int J Mol Sci 24:12989, 2023). This paper should be cited and discussed in the manuscript.

Reply: We appreciate the reviewer's suggestion. This paper is relevant to our research and required to be discussed in the text. We cited this paper (Ref. 75) and revised the discussion section on Page 15

Previous works on *il11* across animal species and tissues have revealed seemingly conflicting results, as *il11* is known to have both regenerative and fibrotic properties^{18, 19, 20, 21, 22, 23, 30, 70, 71, 72, 73, 74, 75}. Regarding the hearts, antifibrotic/cardioprotective and fibrotic roles of *il11* have been reported^{21, 22, 23, 30, 74, 75}. Both *il11ra* mutant (loss-of-function for IL11 signaling)²¹ and *il11OE* (gain-of-function in this work and Ref. 75)⁷⁵ models exhibited cardiac fibrosis phenotypes. But their underlying mechanisms of fibrosis differ fundamentally. *il11ra* mutant hearts lost the regenerative ability, leading to impaired function. This can cause cardiac dysfunction, thereby inducing pathological fibrosis as a compensatory mechanism. In contrast, *il11aOE* activates epicardial cells to induce ECM remodeling during the initial phase to establish a regenerative niche. However, prolonged activation of epicardium by *il11* results in constant generation of cardiac fibroblasts, ultimately leading to cardiac fibrosis. Similar outcomes regarding regenerative pathways are also reported by the Hippo pathway. While *Yap1* conditional mutation in CMs of neonatal mice exhibits fibrotic hearts⁷⁶, ectopic activation of *Yap1* in cardiac fibroblasts also induces fibrosis^{77, 78, 79, 80}. Therefore, duration of regenerative factor treatment and their target cells should be considered to achieve successful heart repair without adverse outcomes.

Reviewer #1 (Remarks to the Author):

1. The Shin et al. manuscript comes in a somewhat improved version after some of the concerns were addressed, however, some of my major concerns have unfortunately not been addressed with the necessary rigor.

In mammals, it was shown that IL11 promotes fibrosis (e.g. Schafer et al, 2017). In zebrafish, it was shown that IL11 is essential for tissue regeneration, limiting fibrosis (e.g. Tsujioka et al., 2017, Allanki et al., 2021). While there are several possibilities why that could be the case, including postulated differences resulting from evolution (Cook JofCTR 2023), the specific strength of the work from Shin et al. lies in its gain of function approach in zebrafish potentially being able to contribute to help clarifying this controversy.

The central assertion of the manuscript is that IL11 overexpression in zebrafish leads to fibrosis, akin to observations in mammalian models. However, this claim remains to lack sufficient empirical support. While the authors have refined their language to some extent to clarify that regeneration involves transient extracellular matrix (ECM) deposition which is obviously different from pathological fibrosis, they assert that cardiac IL11 overexpression results in fibrosis without adequately demonstrating the upregulation of fibrosis markers widely recognized in mammalian studies and presented in the zebrafish knockout study (e.g. Allanki et al. Sci Adv 2021).

In the referenced study, a loss of IL11 signalling was associated with a broad activation of a fibrotic gene program, including essential regulators of fibrosis such as TGF1/2, SOX9, EGR1, and ECM cross-linking enzymes like LOX, which distinguish fibrotic from regenerative ECM deposition. In addition it was shown that loss of IL11 signalling leads to a differentiation of ACTA2+ myofibroblasts from both the epicardial and endocardial layers. Notably, myofibroblast differentiation was shown to be mitigated by reactivating IL11 signalling in a cell type-specific manner, affecting only endocardial but not epicardial cells.

Hence in my previous review, I recommended directly assessing the fibrosis-specific markers to clearly differentiate between regenerative ECM remodelling and fibrosis. However, the authors have decided to limit their analysis and to only present a limited range of ECM genes, including various collagens (and FN1) that are also upregulated during regeneration, and thus are not unique to pathological types of fibrosis.

Furthermore, the origin of the observed ACTA2+ cells in the IL11 overexpression animals at late time points remains unclear. The authors have not been able to provide evidence that these cells are derived from lineages representing the canonical progenitors of cardiac fibrosis seen in mammals. The authors performed additional immunostaining to observe that these ACTA2+ cells were not cardiomyocytes, not endothelial cells, and not epicardial derived cells. Hence, they could as well be other ACTA2+ cell types including SMC, or, based on their anatomical location (in between cortical and trabecular cardiomyocyte layers) they could be primordial layer cardiomyocytes, which were previously shown to be negative for sarcomeric structures (Tadesse-Tsedek et al, Dev Biol 2021). This ambiguity could be resolved by co-staining with embCMHC or MHC (MF20) or HABC. If indeed these ACTA2+ cells are primordial layer CMs, it would mean that IL11 overexpression induces CM dedifferentiation, which is a regenerative response – quite the opposite of fibrosis. In line with this, the authors have already shown in Fig S2 that IL11 OE hearts indeed display increased CM dedifferentiation. Hence, I strongly recommend the authors check their bulk RNA-seq both at 7 dpt and 7 mpt for the fibrotic markers mentioned above, and CM dediff markers, e.g. those published in Tadesse-Tsedek et al 2021. In addition, in Fig S9e, the authors

show increased ACTA2+ cells in IL11 OE hearts. They look more like vessel-associated SMCs. A co-staining with endothelial markers at this time point would be helpful.

From my perspective further characterizing these ACTA2+ cells and the fibrotic gene program appears paramount in calling it a fibrotic response.

In summary, I believe that the manuscript's central claim is not sufficiently substantiated by the necessary data. Substantial further investigation into the specific markers of fibrosis appears crucial to robustly support the conclusions drawn. Alternatively, the authors should substantially moderate their claim.

Reply: We appreciate the reviewer for the thoughtful comments. We also apologize for insufficient explanation to clarify fibrosis caused by *il11a* overexpression (OE). In the revised manuscript, we have added new data clarifying fibrotic mechanisms and address the reviewer's concerns.

1) *il11a*-induced cardiac fibrosis

In our revised manuscript, we have added new data elucidating the mechanisms underlying *il11a*-induced fibrosis. We demonstrate that *il11a*OE gradually induces the cardiac fibroblast layer expressing Periostin (Postn) within the cortical layer. Previous studies showed that Postn is strongly expressed in cardiac fibroblasts contributing fibrosis in injured zebrafish and mammalian hearts (Sanchez-Iranzo H *et al*, *PNAS*, 2018; Fu *et al.*, *JCI*, 2018). While Postn+ cardiac fibroblast is undetectable in *il11a*OE at 10 days post-treatment (dpt), we found that *il11a*OE induction leads to the emergence of Postn+ cell layer within the cortical area (**Fig. 4d, e**). Our quantification demonstrated that Postn+ cell area gradually expands over time, matching *il11a*-induced fibrotic outcome.

We next address the cellular origins of these Postn+ cardiac fibroblasts. Since our *il11a*OE model requires to use the cardiomyocyte (CM)-inducible *CreER* line (*cmhc2:CreER*) for the overexpression experiment, the lineage tracing experiment for epicardial cells (*tcf21:CreER* and *ubb:switch* lines) is not feasible due to using two *CreER* lines. It's important to note that tamoxifen treatment will induce recombination of ubiquitously expressed floxed reporter allele in both epicardial and CMs, thus labeling both cell types. As an alternative approach, we assessed epicardial marker expression with Postn+ cells and find that ~66% of Postn+ cells are co-labelled with *tcf21*+ epicardial marker (**Fig. 4f, g**). These findings suggest that Postn+ cardiac fibroblasts originate from epicardial cells. Overall, our new results identify that the fibrotic effect of *il11* is caused by prolonged activation of epicardium, leading to the generation of Postn+ cardiac fibroblasts and contributing to fibrosis.

These new results are included in Figure 4.

We have added new text on Page 10.

In regenerating zebrafish hearts, cardiac fibrosis is largely mediated by epicardium-derived fibroblasts that appear in the cortical layer following injury ^{6, 13, 63, 64}. Given that the fibrosis driven by *il11a* is predominantly observed in this cortical layer, we hypothesize that prolonged *il11a* exposure can activate the epicardium constantly, leading to sustained cardiac fibroblast production and subsequent fibrosis. Supporting this notion, while Periostin+ (Postn+) cells are undetectable in the 10 dpt *il11a*OE hearts, *il11a*

induction led to the emergence of Postn⁺ fibroblasts in the cortical layer at 30 dpt without an injury (**Fig. 4d, e**). These Postn⁺ fibroblasts are gradually expanded at 3 mpt and dispersed across the cortical layer, where collagens are extensively deposited. Next, we examined whether epicardial cells can give rise to the Postn⁺ fibroblasts. Notably, 66% Postn⁺ cells are colocalized with *tcf21*:mCherry⁺ epicardial cells in *il11a*OE at 30 dpt while there is no Postn expression in epicardial cells of control hearts (**Fig. 4f, g**). Thus, our results indicate that *il11a*-induced fibrosis is mediated by epicardial-derived Postn⁺ cardiac fibroblasts.

2) RNA-seq analysis to examine regenerative and fibrotic gene expression

To address reviewer's concern regarding RNA-seq analysis, we have analyzed our 7 days post-treatment (dpt) and 7 months post-treatment (mpt) transcriptome profiles with broader ranges of regeneration and fibrosis genes. Regarding regeneration-associated genes, we focus on regenerative extracellular matrix (ECMs), embryonic CM genes (CM dedifferentiation markers), and cell cycle-related genes. These include *lepb*, *fn1b*, *col12a1b*, *hmmr*, *plk1*, *cdk1*, *ccnb1*, *ccnb2*, *mki67*, *nppa*, *nppb*, *desma*, and *ankrd1a*. For fibrosis-associated genes, we examine various collagens and essential fibrosis genes, including *sox9a/b*, *dcn*, *lox* enzymes (ECM linking enzymes), fibroblast activation genes, ECM maturation genes, and members of transforming growth factor β (*tgfb*). Our analysis reveals upregulation of regenerative genes in 7 dpt *il11a*OE heart, while fibrosis-associated genes exhibit elevated expression at 7 mpt, but not 7 dpt. We also find a modest, but not robust, upregulation of one of *tgfb* genes. This *tgfb* result aligns with the previous finding that *Il11* was uncovered as a downstream effector of Tgfb signaling for cardiac fibrosis. Overall, our new data demonstrate the upregulation of fibrosis markers widely recognized in mammalian and zebrafish studies. These results are shown in **Fig. 4c** and **Supplementary Fig. 7a**.

Note that we do not include *egr1* as a fibrotic gene due to its diverse functions in hearts. Egr1 is known to play roles in autophagy, angiogenesis, and cardiac hypertrophy, indicating that further validation is required to categorize Egr1 as a fibrotic gene. (References: Khachigian, 2021, JAHA; Billah et al., Heart Lung Circ., 2023)

These new results are included in **Fig. 4c** and **Supplementary Fig. 7a**.

We have added new text on Pages 9 and 10.

To assess the induction of fibrosis-associated genes by prolonged *il11a*OE, we performed RNA-seq analysis with 7 mpt control and *il11a*OE hearts, comparing them with 7 dpt transcriptome datasets. The border zone of the injured hearts exhibits high enrichment of embryonic CM genes, including *nppa*, *nppb*, *desma*, and *ankrd1a* ^{51, 52, 53}. These embryonic genes, in addition to cell cycle and regenerative ECM genes (**Supplementary Fig. 2c and 7a**), are upregulated early at 7 dpt after *il11a* induction, suggesting the prompt regenerative effect of *il11a* in hearts (**Supplementary Fig. 7a**). By contrast, our transcriptome analysis

revealed a significant upregulation of pathological fibrotic and cardiac fibroblast marker genes, including *col1a1a/b*, *col5a1*, *postnb*, *sox9a/b*, *dcn*, fibroblast activation genes (*cthrc1a*, *thbs4b*), ECM-linking genes (various Lysyl oxidase (LOX) enzymes), ECM maturation genes (*mgp*, *sparc*) and in 7 mpt *il11a*OE hearts, but not 7 dpt (**Fig. 4c, Supplementary Fig. 7a, b, and Supplementary Data 2 and Table 2**)^{13, 54, 55, 56, 57, 58, 59, 60, 61, 62}. Extended *il11a* induction moderately, but not robustly, upregulates one of two *transforming growth factor β 1* (*tgfb1*) homologs (**Supplementary Fig. 7a**), aligning with the previous finding that *il11* plays roles as a downstream effector of *Tgfb1* for cardiac fibrosis²². These results highlight the dual functions of *il11* as a regenerative factor with more immediate effects and a fibrotic factor, presumably, with indirect consequences.

3) Clarification of ACTA2 expressing cells

ACTA2 is a well-characterized myofibroblast marker in mammalian hearts. Since *il11a* induction produces additional ACTA2⁺ layers within the cortical area, we previously hypothesized that *il11a*OE activates epicardium to constantly give rise to ACTA2⁺ myofibroblasts, leading to cardiac fibrosis. As the reviewer indicated, ACTA2 can label multiple cell types, including dedifferentiating cardiomyocytes (CMs), vascular smooth muscle cells (VSMCs), and myofibroblasts. Therefore, in this revised manuscript, we perform immunostaining ACTA2 with endothelial (*flil1:EGFP*) and CM (myosin heavy chain, MHC) markers. Our analysis found that ACTA2⁺ cells induced by *il11a*OE are indeed co-labelled with MHC (**Supplementary Fig. 8**). Agreeing with the reviewer's comment, these results suggest that *il11a*-induced ACTA2⁺ layer is dedifferentiating primordial CM layer. These results offer another evidence supporting the regenerative effect of *il11a*. As shown in response 1), we found that Postn⁺, rather than ACTA2⁺, cardiac fibroblasts mediate fibrotic effect of *il11*.

These results are included in **Supplementary Fig. 8**.

We have revised our text on Pages 11, 12.

ACTA2, smooth muscle alpha (α)-2 actin, is used as a biomarker for the injury-induced myofibroblasts in mammalian hearts^{65, 66}, raising a question of ACTA2 expression in response to *il11a*OE. In addition to myofibroblasts, ACTA2 also labels vascular smooth muscle cells (VSMCs) and dedifferentiating CMs^{51, 67}. While circular ACTA2⁺ cell layers were observed in the cortical layers of both control and *il11a*OE hearts, *il11a* induced the additional ACTA2⁺ cells in the cortical layer without an injury at 30 dpt (**Supplementary Fig. 8a-e**). To discriminate ACTA2-expressing cells, we examined their location and colocalization with blood vessels and CM markers. In control hearts, most ACTA2⁺ cells were found encircling *flil1a:EGFP*⁺ coronary vessels within the outer cortical layer of ventricles, an indicative of VSMC

(**Supplementary Fig. 8c**). Although circular ACTA2⁺ VSMC layers were also evident in *il11a*OE hearts, we found that *il11a*OE hearts additionally display a thick and linear ACTA2⁺ layer in the deeper cortical layer (**Supplementary Fig. 8c**). These ACTA2⁺ layers are not closely associated with *fli1a:EGFP*⁺ cells, indicating non-VSMC cell types (**Supplementary Fig. 8d**). While the majority of these ACTA2⁺ cells lack α -actinin CM signature (**Supplementary Fig. 8e**), they co-express a myosin heavy chain (MHC) CM gene (**Supplementary Fig. 8f-h**). Based on the anatomical location of the primordial CM layer that gives rise to trabecular CMs during development⁶⁸, our results indicate that *il11a*-induced ACTA2⁺ layers in the deeper cortical layer are likely dedifferentiating CMs.

In addition, all related to the above.

2. In line 206 and 207 the authors write: “ACTA2 is used as a marker for vascular smooth muscle cells (VSMCs) and dedifferentiating CMs in addition to cardiac fibroblasts” – That is not correct. ACTA2 is not a marker for cardiac fibroblasts but for myofibroblasts. Quiescent cardiac fibroblasts do not express SMA/ACTA2 under physiological conditions. Furthermore, SMA/ACTA2 is not exclusive to fibroblast derived myofibroblasts but is also activated in other lineages (eg. Endothelial derived myofibroblasts). To discriminate myofibroblasts from SMCs, additional coexpression analysis for MYLK or SOX9 might help.

We appreciate the reviewer for this suggestion. As described above (3) Clarification of ACTA2 expressing cells), we demonstrated that ACTA2⁺ cells are VSMCs and dedifferentiating primordial CMs rather than myofibroblasts in *il11a*OE hearts.

3. Related to the above, the authors claim in Line 251-252 of the revised manuscript that “*Our data indicate that, while il11a induction gradually accumulates collagen in uninjured hearts, its fibrotic function is accelerated in the presence of injury.*” – This claim ignores that genetic LOF data has been established in zebrafish providing evidence that *il11a* is essential for regeneration and limits fibrosis after injury. In addition, the authors show that COL12 reporter activation (Fig S9) and CM dedifferentiation (Fig S2) – both established regenerative responses – are exacerbated in the IL11 OE hearts. And the authors showed an increasing vascularization (also a regenerative response) of the IL11 OE ventricles without injury (Fig 3a-d). In line with this, the increased ACTA2⁺ cells in Fig S9e could be SMCs, which would imply an increased revascularization after injury as well. All these data (COL12 reporter, CM dediff, and revasc) suggests that these IL11 OE hearts are stuck in a continuous regenerative program possibly inducing progressive heart failure. In its current form, the manuscript does not contain sufficient data allowing to discriminate between these conditions.

For our statement regarding the acceleration of fibrosis in injured hearts, our intention is not to ignore the previous finding, instead reporting our observation: while persistent fibrosis is observed in 30 days post-amputation (dpa) injured hearts, cardiac fibrosis is not induced in *il11a*OE hearts at 30 dpt. However, we agree that our statement requires to be revised. Our new results revealed the presence of Postn⁺ cardiac fibroblasts in injured hearts, suggesting fibrotic function of *il11a* in injured hearts.

These new results are included in **Supplementary Fig. 9**.

We have added new text on Page 13.

... Furthermore, ectopic *il11a* induction led to the persistence of *col12a1b*⁺ EPCs and fibrotic tissues at the injury site in cryoinjured hearts (**Supplementary Fig. 10a-d**). Similar to *il11a*OE uninjured hearts, we observed prominent Postn⁺ layers at the wound area of *il11a*OE hearts, suggesting the existence of cardiac fibroblasts induced by *il11a*OE. (**Supplementary Fig. 10e, f**). Overall, these results indicate regenerative and fibrotic functions of *il11a* in injured hearts.

4. 322-326 *“Both il11ra mutant (loss-of-function for IL11 signaling) and il11OE (gain-of-function in this work and Ref. 75) models exhibited cardiac fibrosis phenotypes. But their underlying mechanisms of fibrosis differ fundamentally. IL11ra mutant hearts lost the regenerative ability, leading to impaired function. This can cause cardiac dysfunction, thereby inducing pathological fibrosis as a compensatory mechanism.”*

- According to the reference, IL11 mutants show a robust activation of a broad fibrotic gene program, as early as 96 hours post cardiac injury. Hence, fibrosis in this case is very unlikely to be attributed to heart failure resulting from impaired regeneration. Furthermore, no impaired heart function or even heart failure was reported in the cited work. Notably, the same argument falls back on this manuscript as the ECM deposition described here as “fibrosis after IL11 overexpression” could be a consequence from the globally and continuously dedifferentiating cardiomyocytes “leading to impaired heart function. This can cause cardiac dysfunction, thereby inducing pathological fibrosis as a compensatory mechanism.”

– Given the extended period of time it takes IL11OE in zebrafish to induce significant ECM deposition compared to *il11ra* LOF induced fibrosis, the authors should possibly more deeply think about this argument in the light of their own conclusions.

We appreciate the reviewer for thoughtful comment. Our data demonstrate that a regenerative paracrine factor can target multiple cell types, and for successful regeneration their expressing location and duration are also crucial. Previous loss-of-function study identified the *il11*'s antifibrotic function in endocardium to prevent endothelial-to-mesenchymal transition (EndoMT). Our gain-of-function study identified the *il11*'s regenerative functions in CMs and epicardium. However, prolonged induction can lead to aberrant outcomes, such as fibrosis. In a revised manuscript, we have revised our discussion to emphasize the pleiotropic effect of *il11* depending on target cells and duration.

We have revised the text on Page 16.

Previous works on *il11* across animal species and tissues have revealed seemingly conflicting

results, as *il11* is known to have both regenerative and fibrotic properties ^{18, 19, 20, 21, 22, 23, 30, 80, 81, 82, 83, 84, 85}. Regarding the hearts, cardioprotective and fibrotic roles of *Il11* have been reported ^{21, 22, 23, 30, 84, 85}. Both *il11ra* mutant (loss-of-function for IL11 signaling) ²¹ and *il11OE* (gain-of-function in this work and Ref. 85)⁸⁵ models exhibited cardiac fibrosis phenotypes. Loss-of-function study demonstrated that *il11ra* mutants exhibit an elevated number of myofibroblasts derived from endocardium and epicardium, leading to defective resolution of cardiac fibrosis. Tissue-specific rescue experiment of *il11ra* mitigates endocardium-derived myofibroblast differentiation, but not epicardium, suggesting the key roles of *il11* signaling in the endocardium for endothelial-to-mesenchymal transition (EndoMT). Our gain-of-function study revealed that *il11a*-mediated fibrotic effect is indirect through prolonged activation of epicardium. *il11*-induced epicardial activation is crucial to establish a regenerative niche at the initial phase of regeneration. However, persistent epicardium activation by *il11* results in continuous generation of cardiac fibroblasts, ultimately leading to cardiac fibrosis. These findings can explain controversies of *il11* signaling by highlighting pleiotropic function of *il11* depending on the location and duration of its expression.

Reviewer #2 (Remarks to the Author):

The authors have addressed all the specific issues that we have raised in a satisfactory manner. The overall quality of the data is high and the conclusions drawn are sufficiently backed by data.

However, in our review we pointed out that we thought that additional mechanistic insights might be needed to increase the overall novelty of the findings. Unfortunately, the authors were not able to add much along those lines. We appreciate that the authors made efforts; unfortunately, the first suggestion we made (overexpression in epicardium or endocardium) could not be done using the existing tools, since the Cre responder is not expressed in these cells. The second point was only met with a rather general discussion. For the third, the authors do provide additional evidence that the IL OE plus ERK inhibition does slightly increase CM numbers over unperturbed (vehicle treated wild-type) regeneration. Yet, as a potentially “therapeutic” approach this is not very impressive.

Thus, whether the paper presents enough novelty for Nat Comm is for the editors to decide. The existing data are certainly solid and the conclusions well supported.

We appreciate the reviewer for his/her overall support and critical comments on our manuscript.

Regarding *il11a* overexpression in other cell types, we encountered the technical limitations with our current *il11a* floxed allele not expressing in epicardial and endocardial cells. I agree that determining the effect of *il11a* based on expressing cell types is important, but also maintain that the main goal and chief

impact of the current manuscript is to demonstrate the dual functions of *il11a* for heart repair and how to harness its regenerative effect. We are establishing a new transgenic line that will allow us to induce *il11a* in other cell types. We expect that we will address the reviewer's concern as a separate study to determine the effects of *il11a* expressing cell types.

Regarding comparison of loss-of-function study, we have incorporated new data elucidating the mechanisms of *il11a*'s fibrotic effect and revised our discussion accordingly (please see the response to Reviewer 1).

Regarding the effect of combinatorial treatment, our experiment demonstrated that ERK inhibition at the later time point of injured hearts is unlikely to influence *il11a*-stimulated CM proliferation, but it significantly mitigates the cardiac fibrosis induced by chronic *il11a* induction. These results clearly provide proof-of-concept that combinatorial approach is effective to heart repair. As we described in the discussion, this approach offers a promising and novel strategy for utilizing a secreted paracrine factor for heart repair. In addition to transient induction of a regenerative factor in the field of heart regeneration, our study can introduce a new and promising methodology for heart repair. To improve the efficiency of our approach, further optimization is needed by examining the different doses, durations of the drug, and delivery methods. This will require a wide range of experimental designs, which demand substantial effort and time. We have initiated new experiments with a mouse model to establish an effective, accurate, and reliable strategy toward therapeutic application. This study would extend significantly beyond what we are attempting to accomplish in this first manuscript and would constitute separate (albeit high-impact) studies.

I hope the reviewer will agree that we have made our best efforts in response to all the comments we received from reviewers. We have a high level of confidence that our study will provide incentive for the field of heart regeneration to tackle the sophisticated questions s/he has asked.

Reviewer #3 (Remarks to the Author):

I appreciate the authors' effort to address my comments. I am still awkward to agree with the conclusion in the revised manuscript: "Collectively, these results demonstrate that *il11a* activation can bypass the injury signal to induce majorly myocardial hyperplasia, highlighting the mitogenic effect of *il11a* in zebrafish hearts." (p. 7, lines 139-141.) To me, the data shown in Figure 2 suggests a very moderate proliferative effect of IL11 overexpression, and "majorly" doesn't seem to fit with the data. Because of this moderate phenotype, I also feel that the authors should consider reducing the tone about the mitogenic role of IL11 in other part of the manuscript.

We appreciate the reviewer for the comments. As the reviewer recommended, we have toned down our statement about mitogenic role.

The revised parts are:

Page 4:

***il11a* induction can promote cardiomyocyte proliferation in uninjured hearts**

Page 5:

... muscle cells upon tamoxifen treatment in the absence of injury (**Fig. 2a**). Strikingly,

-> ... muscle cells upon tamoxifen treatment in the absence of injury (**Fig. 2a**). **Notably,**

..., providing evidence for the mitogenic roles of *il11a* on CMs.

-> ..., **suggesting that *il11a*OE is sufficient to trigger CM proliferation.**

Page 7:

... Collectively, these results demonstrate that *il11a* activation can bypass the injury signal to induce majorly myocardial hyperplasia, highlighting the mitogenic effect of *il11a* in zebrafish hearts.

-> ... Collectively, these results **indicate that *il11a* activation can lead to notable myocardial hyperplasia, highlighting its regenerative effect in zebrafish hearts.**

Page 12:

... confirming its mitogenic effect on injured hearts. ...

-> ... confirming its **regenerative** effect on injured hearts. ...

Page 14:

... We demonstrate that *il11a* induction in the uninjured heart can bypass cardiac injury cues to stimulate regeneration programs, ...

-> ... We demonstrate that *il11a* induction in the uninjured heart can stimulate regeneration programs, ...

REVIEWERS' COMMENTS

Reviewer #1 (Remarks to the Author):

In this revised version, Shin et al. provide additional data intending to add experimental support to previous claims and to correct incorrect statements in the previous version.

In short:

Background: Shin et al. use an overexpression line to dissect possible roles for Interleukin-11 (IL-11), an interleukin-6 family cytokine previously linked to regeneration and fibrosis in various experimental systems, most importantly in mouse, rat, and zebrafish hearts. While genetic loss- and gain-of-function data in mice revealed a profibrotic function for IL-11 in mice (by promoting ACTA2+ myofibroblast differentiation), genetic data in zebrafish revealed a crucial function for IL-11 to promote regeneration (limiting myofibroblast differentiation). In rats, systemic IL-11 administration showed cardioprotective effects, and rhIL-11 is an FDA-approved drug.

Study: The authors use a gain-of-function approach by conditionally overexpressing IL-11 in cardiomyocytes of zebrafish to shed light on the question of whether excessive IL-11 promotes regeneration or fibrosis. Previously, the authors showed that aspects of a regenerative response, including hypervascularization and increased cardiomyocyte proliferation, are induced by IL-11 overexpression at early stages close to the activation of the IL-11 overexpression. These findings were in line with previous reports that IL-11 acts as a pro-regenerative cytokine in zebrafish and other models. The authors further claimed that prolonged expression results in fibrosis based on the identification of ACTA2+ cells, which were interpreted as myofibroblasts (a hallmark of fibrosis), and ECM deposition. However, the origin and nature of the ACTA2+ cells remained unclear, and the detected ECM deposition cannot be interpreted as maladaptive remodeling without further evidence, as ECM deposition is also a hallmark of regeneration.

In this revised version of the manuscript, the authors now partially address previous concerns and substantiate some of the claims.

New results:

The authors now add data showing that IL-11 overexpression results in delayed Periostin expression in tcf21 positive cells, a marker for activated cells in the context of regeneration and fibrosis in zebrafish (Figure 4 e, d, f, g). As with the previous analysis of other markers, Periostin is not a fibrosis-exclusive marker but is also expressed during regeneration, leaving this analysis on its own insufficient to draw strong conclusions. This problem becomes bigger as the canonical hallmark of cardiac fibrosis, the differentiation of ACTA2+ myofibroblasts from cardiac stromal cells, could not be validated during revisions but turned out to be dedifferentiating cardiomyocytes.

However, the extended period of time of 30 days after activation of the overexpression suggests that Periostin expression in this case (30+ days) is not a direct consequence of IL-11 expression but rather of secondary effects, supporting the notion that Periostin expression can be regarded as a consequence of maladaptive cardiac remodeling.

Hence, the authors now use RNAseq data to investigate the dynamics of regeneration and fibrosis in their system using a fibrotic and regenerative marker gene set at different stages after IL-11 activation. This experiment now shows that even seven days after transgene activation,

none of the fibrotic markers is activated through IL-11 overexpression, but rather the regeneration-related gene set shows activation. Only at seven months post-transgene activation (7 mpt), enhanced gene expression of fibrotic genes was detected, including periostin expression.

Taken together, while previously published results for early and possibly direct activation of regenerative genes could be established, the prolonged time to induce fibrotic gene set expression rather speaks for an indirect effect of IL-11, possibly caused by organ failure. The causes could result from dedifferentiation and excessive growth of the myocardial layer, but the analysis is speculative at this point.

I am still hesitant to attribute fibrosis directly to IL-11 overexpression in this model exclusively based on the activation of a fibrotic gene set and Periostin expression only after 30 days to three months, with no detectable emergence of myofibroblasts and only after clear dedifferentiation of cardiomyocytes occurred. In my view, the authors should clearly discuss the possibility that this is a secondary, indirect effect, as now also included in the text section related to point 4.

The authors now clearly state that “Our gain-of-function study revealed that il11a-mediated fibrotic effect is indirect through prolonged activation of epicardium” in the revised text. I believe this is a fair interpretation of the results and also compatible with the observed gene expression and Periostin activation, and should be the general interpretation of the work rather than a direct consequence. However, this remains inconsistent in the current version of the manuscript, e.g., Line 215 (“Given that the fibrosis driven by il11a...”) should be “Given that the fibrosis observed in il11a overexpressing hearts” or Line 225: “il11a-induced fibrosis” should be “fibrosis resulting from ectopic il11a expression” and more.

Of note: The authors write “Tissue-specific rescue experiment of il11ra mitigates endocardium-derived myofibroblast differentiation, but not epicardium, suggesting the key roles of il11 signaling in the endocardium for endothelial-to-mesenchymal transition (EndoMT).” To my understanding, in the cited experiment, il11ra was exclusively rescued in the endocardium, which sufficiently explains why myofibroblast differentiation was only affected in these cells but not in the epicardium. The fact that a global loss of il11ra leads to an increase in myofibroblast differentiation in both the endocardium and epicardium strongly suggests key roles in both cell types.

Reply: We appreciate the reviewer for his/her overall support and critical comments on our manuscript. As the reviewer recommended, we have edited out statement in page 10 and 11.

Page 10

Given that the fibrosis driven by *il11a* is ...

-> Given that the fibrosis driven by *il11a* **overexpressing hearts** is

Page 11

Thus, our results indicate that *il11a*-induced fibrosis is ...

-> Thus, our results indicate that **fibrosis resulting from ectopic *il11a* expression** is ...

Reviewer #2 (Remarks to the Author):

In my response to the revisions I had pointed out that I had found it somewhat disappointing that

the authors had not been able to add more data to the manuscript that would more clearly illustrate its advance over existing literature. With this second revision they have now expanded on the issue of how their finding that il11a overexpression induces fibrosis fits with the published results that il11ra mutants likewise present with fibrosis. I also appreciate the more detailed discussion and requests for additional data that reviewer 1 has had with the authors along those lines.

Overall, I find that the paper is further strengthened. The main message, which in my view is that the complicated pro- and anti-regenerative roles of il11 are not only seen in the poorly regenerative mammalian hearts, but also in highly regenerative zebrafish, is worth reporting in Nat Comms.

Reply: We appreciate the reviewer for his/her overall support and critical comments on our manuscript.

Reviewer #3 (Remarks to the Author):

I appreciate the revision, which seems more accurate in describing the results.

Reply: We appreciate the reviewer for his/her overall support and critical comments on our manuscript.